# Study on the change characteristics and market driving factors of residential land price in the Beijing Tianjin Hebei urban agglomeration, China

Pengfei An[1,2], Can Li[1,2]*, Yajing Duan[3], Jingfeng Ge[1,2], Xiaomiao Feng[4]*

**1** College of Resource and Environmental Science, Hebei Normal University, Shijiazhuang, China,
**2** Hebei Key Laboratory of Environmental Change and Ecological Construction, Shijiazhuang, China,
**3** Shijiazhuang Public Resources Trading & Service Center, Shijiazhuang, China, **4** College of Resources and Environment, Shijiazhuang University, Shijiazhuang, Hebei Province, China

* lican3008@126.com (CL); fengxm0310@126.com (XF)

## Abstract

Land prices are the key problem of urban land management, with prices of residential land being the most sensitive and the strongest social reflection among the different land types. Exploring spatial and temporal variation of residential land prices and the effect of land market factors on residential land prices can help the government formulate targeted regulations and policies. This study analyzes the spatial and temporal evolution of residential land prices and the factors influencing the land market in the Beijing-Tianjin-Hebei region based on land transaction data from 2014–2017 using exploratory spatial data analysis (ESDA) and a geographically weighted regression (GWR) model. The results show the following: ① Residential land prices in Beijing and Tianjin are significantly higher than those in other regions, while Zhangjiakou, Chengde, and western mountainous areas have the lowest residential land prices. Over time, a development trend of residential land price polycentricity gradually emerged, and the locational correlation has gradually increased. ② Under the influence of the land finance model of local governments in China, three factors, namely, the land stock utilization rate, revenue from residential land transfers, and the growth of residential land transaction areas, have significantly contributed to the increase in residential land prices. ③ Under the land market supply and demand mechanism and government management, four indicators, namely, the land supply rate, the per capita residential land supply area, the degree of marketization of the residential land supply, and the frequency of residential land transactions, have suppressed the rise in residential land prices. ④ The overall effect of land market factors on residential land prices in the central and northern regions of Beijing, Tianjin and Hebei is stronger than that in the southern regions, which may be related to the more active land market and stricter macromanagement policies in Beijing, Tianjin and surrounding areas.

**Data availability statement:** All relevant data are within the manuscript and its Supporting Information files.

**Funding:** China Natural Science Foundation Thence Foundation (project No.41471090) and Science Foundation of Hebei Normal University (project No. L2021B29). Jingfeng Ge is the owner of China Natural Science Foundation(Fund No:41471090)and Can Li is the owner of Science Foundation of Hebei Normal University Foundation(project No.L2021B29).

**Competing interests:** The authors have declared that no competing interests exist.

## Introduction

Land resources support various social and economic activities, and the extent to which they are reasonably allocated is directly related to the urbanization process, sustainable economic development, and the improvement of people's livelihoods [1]. In recent years, the continuously increasing concentration of China's population in cities [2] has led to increasingly higher requirements for the living environment within cities [3], so the requirements for urban planning and construction [4, 5] and sustainable development [6, 7] are also increasing. With the improvement of China's land utilization efficiency and the reform of the housing system, Chinese real estate has entered a rapid development stage, and the real estate industry has gradually become a pillar of economic growth in all regions. The healthy operation of the residential land market, as the basis of the real estate market, is crucial to this industry's sustainable development, with residential land prices being directly related to coordinated social development and wellbeing and public harmony [8]. Examining the relationship between residential land price change and residential market operation can provide a clearer view of the mechanism connecting residential land market operation to land price changes. Understanding this mechanism can help the government formulate clear regulations and policies to guide the healthy development of the real estate industry.

In the past, scholars focused on analyzing individual drivers of land prices, such as land-use change [9], population structure and distribution [10], the regional environment [11, 12], urban traffic conditions [13, 14], macroeconomic fluctuations [15], national land policies [16–20], and other factors. To assess the factors influencing land prices, these previous scholars considered various variables, such as population, environmental, transportation, social, economic, and policy indicators, providing references for the analysis in this study. However, most of these previous analyses of the factors influencing of land prices were not comprehensive and often neglected the influential roles of land market factors [21]. In addition to the abovementioned drivers, such as population and environmental indicators, factors such as land market supply and demand, the land supply structure, the operational efficiency of the land market, and regulation and policies are increasingly attracting the attention of scholars.

Currently, land market supply and demand affect land prices mainly by affecting the land supply [9, 22, 23]. In analyses of the contributions of factors to urban land prices, Haojing Shen [24] and Ruozhu Zhang [25] confirmed that land supply and demand are the main factors influencing urban land prices. Furthermore, Mostafa [26] and Kershaw P [27] analyzed land market data from Egypt and Australia and proved the correlation between land market supply and demand and the spatial distribution pattern of land price.

In the context of supply-side structural reforms in the land market in China, the behavior of local governments in influencing land prices by constraining the land supply structure has received attention from scholars. Studies have revealed that different land market tiers and supply methods generate different urban land prices. For example, Jinfeng Du [28] argued that since the establishment of the urban land market in China in 1992, the factor that has had the most significant impact on urban land prices is the supply structure of the primary land market under state control, followed by the secondary market supply structure. In addition, different land characteristics have impacts on land prices. For example, Cao Fei [29] and Deng Yu [30] both found that the excavation of the land stock in the city plays a significant role in promoting the rise of urban land prices. Moreover, the implementation of government controls on land supply for different purposes can control real estate prices. Analyzing the methods employed by local governments to promote urbanization in China, JianYong Fan [31] confirmed that an increase in the residential land supply restrains the rise in real estate prices but that this effect weakens with increasing government financial pressure [32].

With the continuous improvement of land markets in various countries, the influences of the degree of land marketization on land allocation and economic development has become a topic of discussion among scholars [33]. Some scholars have argued that an intensification of marketization in the land market could strongly improve urban public facilities and the ecological environment, which would have an indirect, positive effect on urban land prices [34], and that local governments' expectations regarding and dependence on the land market mechanism may significantly impact land prices [35]. Huang Jing [36] and Mou Yan [37] both found that the degree of land marketization is not a direct driver of land price increase and that when both land markets are underdeveloped and land prices are low, local governments increase the proportion of "bidding, auction, and listing" transfers, which leads to an increase in the overall level of urban land prices.

Regarding the selection of models to explore the factors influencing land price, scholars have mostly used econometric approaches, such as multiple linear regression models [38, 39], hedonic price models [40, 41], and hierarchical linear models [42]. With the development of geographic information system (GIS) technology, scholars have introduced the geographically weighted regression (GWR) model into land price-related studies and proven the usefulness of the model. For example, scholars such as Ihlanfeldt [43], KangRich [44], and Harris [45], by establishing relationships between land price and related factors through GWR, have demonstrated that this type of model outperforms OLS regression models and other spatial econometric models such as spatial autoregressive (SAR) models, structural equation models (SEMs), and spatial autoregressive combined (SAC) models. Leveraging the practicality and accuracy of the GWR model, scholars have explored the influences of traffic factors [46], location factors [17], and public service factors [47] on land price and have been able to calculate the contributions of various factors to land price [48] as well as reveal spatial heterogeneity [10, 16]. Many studies have shown that the GWR model has great advantages in investigating the factors influencing land price.

In summary, the literature has proven that urban land market factors drive residential land prices, but two aspects deserve to be explored in depth. First, although the literature addresses a wide range of land market variables affecting land prices, covering land supply and demand, market structure, and the degree of marketization, most studies have analyzed the effects of individual or partial variables, and a comprehensive analysis of the effects of multiple land market factors on land prices is lacking. Changes in residential land prices often result from the combined effects of various factors. Therefore, it is necessary to construct a comprehensive set of land market factors and exhaustively analyze the contribution of each factor to residential land price within the overall context of the land market. Second, most existing studies have used time-series data for analysis and have seldom considered the spatial relationships between land price and land market factors. Against the background of China's current "classified guidance and localized policy" market framework, the relationships between land markets and land prices may differ among different cities. Therefore, analyzing the spatial characteristics of the land market factors that drive residential land prices can provide insight into the effects of land market regulation in different cities and a theoretical basis for the implementation of land classifications and regulations in China.

## Materials and methods

### Research area

The Beijing-Tianjin-Hebei region is the economic center of the north, located between 113°27′~119°50′ E and 36°03′~42°40′ N (Fig 1). It includes two municipalities, Beijing and Tianjin, and 11 prefecture-level cities in Hebei Province, including Shijiazhuang, Tangshan, Baoding, Langfang, Qinhuangdao, Zhangjiakou, Chengde, Cangzhou, Hengshui, Xingtai,

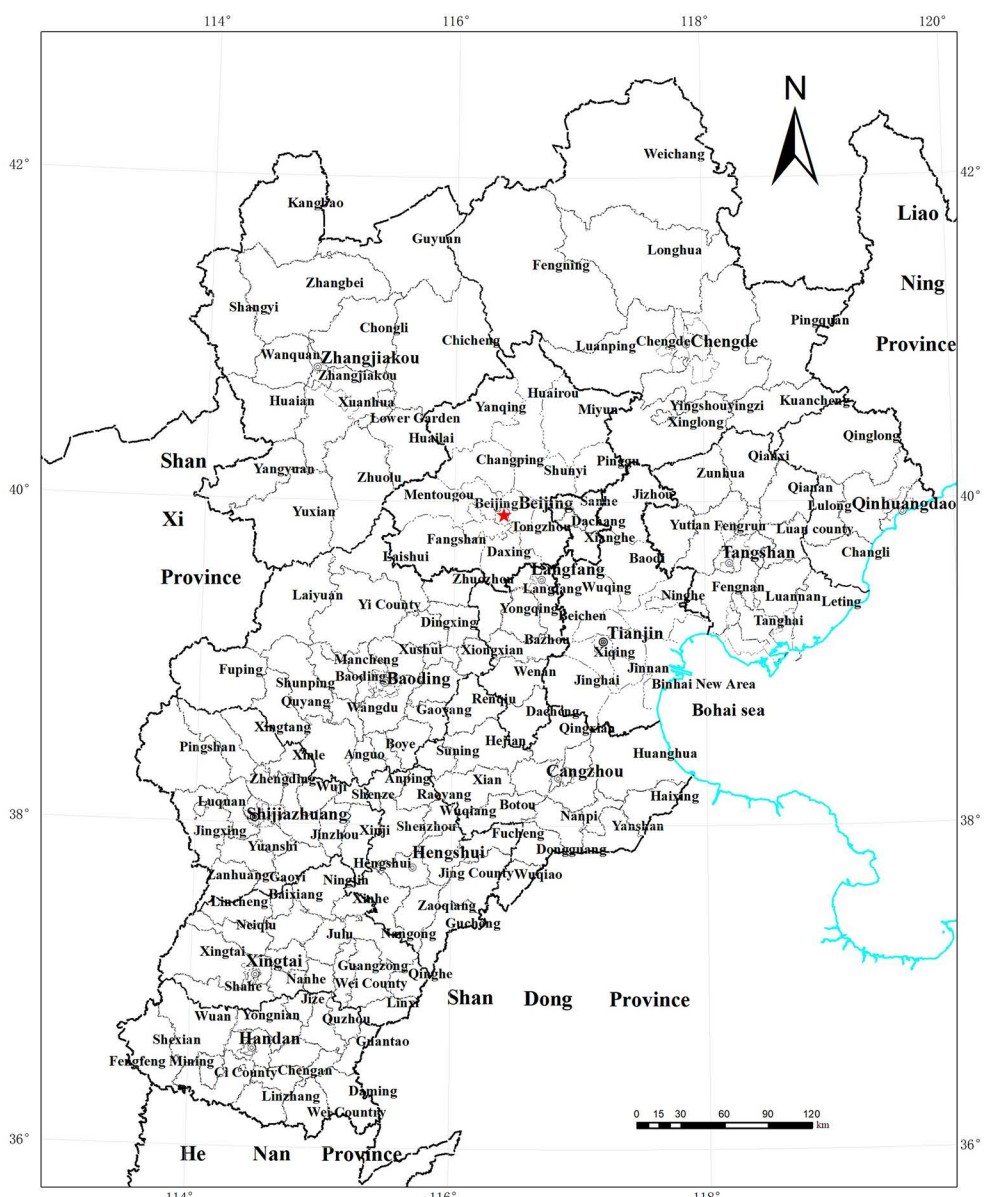

**Fig 1. Location map of the Beijing-Tianjin-Hebei region.**

and Handan, with a total of 200 county-level administrative divisions. The geographical area is 218,000 square kilometers, accounting for 2.3% of the total area of China. As the political, economic, and cultural center of the country, Beijing has significant advantages in terms of its policy orientation, resource allocation, and industrial structure, and its level of economic development and competitiveness is much higher than that of other regions. As an important commercial port, Tianjin is the core area of international shipping in northern China, and its economic development is strongly supported by national policies. However, there is a large gap between the levels of economic development in Hebei Province and the Beijing-Tianjin region. In 2020, the per capita GDP figures for Beijing and Tianjin were 167638 yuan and 90175 yuan, far higher than Hebei's average of 4 yuan. The population distribution in Beijing,

Tianjin, and Hebei Province showed a basic spatial pattern of sparseness in the northern mountainous areas and density in the central and southern plains, especially in Beijing and Tianjin, which are highly concentrated, with population densities (1435 people/km$^2$ and 1305 people/km$^2$) more than three times that of Hebei Province (402 people/km$^2$). These differences in economic development and population density across cities have led to unevenness in urban land price levels. In 2017, overall average residential land prices in Beijing, Tianjin, and Hebei were 38,673, 7,007, and 2,526 yuan/m$^2$, respectively, with land prices in Beijing more than 15 times higher than those of Hebei, representing significant regional differences. To promote the coordinated development of Beijing-Tianjin-Hebei, in 2015, the Chinese government promulgated the "Outline of Beijing-Tianjin-Hebei Cooperative Development Planning"; economic ties within the Beijing-Tianjin-Hebei urban agglomeration have since been increasingly tightened, gradually unlocking synergies and fostering rapid development. In this context, investigating the effect of the residential land market on residential land prices and exploring the path of regional land factor market integration to stimulate the synergistic development of industry and land use in Beijing-Tianjin-Hebei are important tasks.

## Data source

This paper takes 174 counties in Beijing, Tianjin, and Hebei over the period from 2014 to 2017 as the research sample; the data used include residential land prices, land market data, and socioeconomic data for each county administrative unit in Beijing, Tianjin, and Hebei. Among them, residential land prices are based on land transaction data and land market data obtained through the China Land Market Network (http://landchina.mlr.gov.cn/), the websites of the Natural Resources Bureau of each city, and field research. A total of 17,800 land transaction observations were collected, with the information including each parcel's land use, supply mode, rating, area, transaction amount, etc. Land market data include information such as the land supply plan and the total supply of construction land, residential land, stock residential land, bidding, auction, and listing residential land, and affordable housing land as well as residential land transfer income. The socioeconomic data are derived from the China City Statistical Yearbook, Beijing Statistical Yearbook, Tianjin Statistical Yearbook, and Hebei Statistical Yearbook, and include mainly GDP, the disposable income of urban residents, general fiscal budget income, the year-end urban resident population, the price index and other indicators.

## Data preprocessing

This paper aims to reveal the precise distribution pattern of residential land prices and residential land market indicators in Beijing-Tianjin-Hebei. We combined 200 county-level administrative regions in Beijing-Tianjin-Hebei into 174 county-level administrative units. To make the residential land prices comparable across the counties in the region, we standardized the residential land price observations from 2014 to 2017 with December 31, 2017, set as the base date. With regard to the average upper limit of the floor area ratio (FAR) of all the sites offered for sale and the average level of land development in the sample, the former is uniformly set to 2.0 and the latter to the local development standard of " Five-accessible and One-leveling ". The average residential land value for each county was calculated as the arithmetic average based on the adjusted land value of the sites offered for sale in the sample.

## Research methods

**ESDA and kriging interpolation method.** Using panel data on the average residential land price of 174 counties in Beijing-Tianjin-Hebei from 2014–2017 as the research sample, we tested the spatial structure of land price data by means of geostatistical methods and

**Table 1. Comparison of the results of different interpolation methods.**

| Method | variogram model | Prediction error of interpolation results | | | | |
|---|---|---|---|---|---|---|
| | | average error | root mean square error | mean standard error | standard average | standard root mean square |
| 1 | spherical model | -119.8092 | 1826.993 | 1476.940 | 0.0008 | 1.0188 |
| 2 | Gaussian model | -170.8563 | 2086.914 | 1321.064 | -0.0008 | 1.1291 |
| 3 | index model | -132.8017 | 1894.385 | 1446.391 | 0.0106 | 0.9860 |
| 4 | trigonometric function | -162.2924 | 2002.482 | 1349.150 | 0.0167 | 1.0418 |

analyzed the data structure characteristics. Kriging spatial interpolation was used to reveal the features of the spatial distribution of residential land prices.

The kriging interpolation method is based on data obeying a normal distribution, and the structural characteristics of the study data need to be tested and identified before interpolation analysis is performed. We used the Geostatistical Analyst module in ArcGIS10.2 to conduct statistical analysis of the spatial data and test whether the residential land price sample data conform to a normal distribution through histograms and QQ-plots and, where necessary, used logarithmic transformation to obtain normally distributed data meeting the assumption of variability. After that, the ordinary kriging spatial interpolation method was used to perform interpolation analysis on the residential land price data of the 174 counties in Beijing-Tianjin-Hebei. Taking into account the different effects of different variogram fitting models, we performed spatial interpolation for 4 different models (spherical, Gaussian, exponential, and trigonometric function) and obtained different interpolation results. We then compared the error sizes to determine the most reasonable interpolation method. In the comparative analysis, the number of step length groupings was determined to be 12 groups with a step length of 14,035 m; the interpolation results of the four methods are shown in Table 1. The average error and root mean square error of the interpolation results of the spherical model are relatively small, with a standard mean close to 0, an average standard error close to the root mean square, and a standard root mean square close to 1. Therefore, the interpolation effect of the spherical model is relatively ideal.

**Global spatial autocorrelation analysis.** A global spatial autocorrelation analysis was used to analyze the spatial characteristics of the sample across the entire region and to test whether there is a clustering effect in the study data. Moran's I can measure the difference and relevance of the spatial distribution of analysis data by allowing assessment of the similarity of observed values of adjacent spatial locations. The indicator is one of the main methods of measuring spatial autocorrelation. Therefore, we conducted a global Moran's I analysis on the residential land prices of the counties in Beijing-Tianjin-Hebei from 2014 to 2017 as follows:

$$I = \frac{n\sum_{i=1}^{n}\sum_{j=1}^{n}(X_i - \bar{X})(X_j - \bar{X})}{\sum_{i=1}^{n}\sum_{j=1}^{n}Wij\sum_{i=1}^{n}(X_i - \bar{X})^2} = \frac{\sum_{i=1}^{n}\sum_{j\neq 1}^{n}Wij\sum_{i=1}^{n}(X_i - X)(X_j - X)}{S^2\sum_{i=1}^{n}\sum_{j\neq 1}^{n}Wij} \quad (1)$$

where I is the global Moran value, n is the number of research observations, $X_i$ and $X_j$ are the land values of spatial locations *i* and *j*, respectively, and $w_{ij}$ denotes the spatial weights between spatial locations *i* and *j*. When *i* and *j* are neighboring spatial locations, $w_{ij} = 1$; otherwise, $w_{ij} = 0$. Moran's I takes values in the range [−1, 1]. When Moran's I>0, the positive spatial correlation is directly proportional; when Moran's I<0, the spatial difference is inversely proportional; and when Moran's I = 0, the spatial distribution is random. For Moran's I, the standardized statistic Z value is used to test whether the research sample displays spatial

autocorrelation. The p-value reflects the probability that the land price sample data are randomly distributed.

**Selection of market indicators for residential land.** Land prices are a barometer reflecting changes in the land market, and the development of the land market can directly affect the level of land prices. By reviewing many works on the relationship between land prices and land market factors, we found that the latter has a significant impact through four main channels: the land market price discovery mechanism, supply and demand, the land supply structure, and land market activity [49, 50]. Therefore, we selected residential land market indicators representative of the above four factors. The 12 evaluation indicators identified through the preliminary selection are the land price–GDP elasticity coefficient, land price–urban per capita disposable income (UPDI) elasticity coefficient, land price–retail price index (RPI) elasticity coefficient, residential land supply ratio, degree of marketization of residential land supply, utilization rate of stock land [51], residential land supply rate, per capita residential land supply area, proportion of land for affordable housing [52], revenue from residential land transfers [53], residential land transaction frequency, and residential land transaction area [54].

On this basis, we used SPSS26 to evaluate multicollinearity, ruling out indicators with a variance inflation factor (VIF)>10; specifically, we excluded the land price–UPDI elasticity coefficient and the land price–RPI elasticity coefficient, ending with 10 land market indicators that affect residential land prices, as shown in Table 2.

**Geographically weighted regression model.** The GWR model is an improved spatial linear regression model that can effectively explore changes in the relationship or structure between variables due to changes in geographic location [55]. The GWR model can intuitively explain the quantitative relationship of the dependent variable with multiple explanatory variables, superimposing the effects of each explanatory variable and accounting for spatial heterogeneity [56]. This study uses the average residential land price by county as the dependent variable and the ten residential land market indicators as the independent variables. The model calculation formula is as follows:

$$y_i = b_0(u_i, v_i) + \sum_k b_k(u_i, v_i)x_{ik} + e_i, i = 1, 2, \ldots, n \tag{2}$$

where $y_i$ is the dependent variable, $n$ is the number of observations, $k$ is the number of explanatory variables, $X_{ik}$ denotes the $k$-th explanatory variable for the $i$-th observation, $(u_i, v_i)$ are the geographic spatial coordinates of the $i$-th data point, $\beta_0(u_i, v_i)$ is the regression intercept of the $i$-th sample observation, $\beta_k(u_i, v_i)$ is the regression coefficient of the $k$-th explanatory variable on the $i$-th data point, and $\varepsilon_i$ is an independent and identically distributed random error term, which is usually assumed to obey a normal distribution $N(0, \sigma^2)$.

The spatial weights are based on a sample with a certain regional scope and can reflect the degree of association of each observation with the regression points. A Gaussian function is usually adopted:

$$w_{ij} = e^{\frac{1}{2}(\frac{d_{ij}}{b})^2} \tag{3}$$

where $d_{ij}$ denotes the distance between the $j$-th sample point and the regression sample point $i$. $w_{ij}$ is the value of the regression weight of the $j$-th sample point and the $i$-th sample point. $b$ denotes the bandwidth, which is a nonnegative attenuation parameter of the functional relationship between weights and distances.

The determination of bandwidth $b$ in GWR analysis is key to the definition of the spatial weight function [57]. For the selection of the optimal bandwidth, the cross-validation (CV)

**Table 2. Land market indicators affecting residential land prices.**

| Indicator | Formula | Indicator description |
|---|---|---|
| Land price–GDP elasticity coefficient ($X_1$) | $X_1 = \dfrac{\sqrt[j-i]{P_j / P_i} - 1}{\sqrt[j-i]{G_j / G_i} - 1}$ <br><br> Where $X_{(1,2,3)}$ denotes the land price–GDP elasticity coefficient (or land price–UPDI elasticity coefficient/land price–RPI elasticity coefficient), $P$ denotes the average residential land price of each county, $G$ denotes county GDP, and $i,j$ denote years ($j>i$) | Characterizes the relationship between residential land prices and macroeconomic development |
| Residential land supply ratio ($X_2$) | $X_2 = R_{i,j} / T_{i,j}$ <br><br> Where $X_2$ denotes the proportion of the residential land supply area in total land | Reflects the balance of residential land in the market |
| Degree of marketization of residential land supply ($X_3$) | $X_3 = r_{i,j} / R_{i,j}$ <br><br> Where $X_3$ denotes the degree of marketization of the residential land supply, $r_{i,j}$ represents the area of residential land sold by county $i$ through biddings, auctions, and listings in year $j$, and $R_{i,j}$ denotes the total area of residential land supplied by county $i$ in year $j$ | Reflects the structural balance of market land supply |
| Utilization rate of stock land ($X_4$) | $X_4 = C_{i,j} / R_{i,j}$ <br><br> Where $X_4$ denotes the utilization rate of stock land, $C_{i,j}$ denotes the stock land area | Reflects the structural balance of stock and new land use in the market |
| Residential land supply rate ($X_5$) | $X_5 = R_{i,j} / J_{i,j}$ <br><br> Where $X_7$ denotes the residential land supply rate, $J_{i,j}$ denotes the planned area of | Measures the efficiency of land market allocation and the degree of land conservation and land use intensification |
| Per capita residential land supply area ($X_6$) | $X_6 = R_{i,j} / P_{i,j}$ <br><br> Where $X_6$ denotes the per capita residential land supply area, $P_{i,j}$ denotes the urban population in county $i$ in year $j$, and $R_{i,j}$ denotes the total area of residential land | Measures the balance of residential land supply and population scale |
| Proportion of land for affordable housing ($X_7$) | $X_7 = B_{i,j} / R_{i,j}$ <br><br> Where $X_7$ denotes the proportion of land for affordable housing in total land | Measures the supply of and demand for land for affordable housing |
| Revenue from residential land transfers ($X_8$) | $X_8 = N_{i,j} / (N_{i,j} + M_{i,j})$ <br><br> Where $X_8$ denotes revenue from residential land transfers, $N_{i,j}$ denotes revenue from residential land sale in county $i$ in year $j$, and $M_{i,j}$ denotes the local general public budget revenue of county $i$ in year $j$ | Reflects the degree of dependence on land finance and activity intensity in the land market |
| Residential land transaction frequency ($X_9$) | $X_9$ denotes the number of residential land transactions in county $i$ in year $j$ | Allows cross-sectional comparisons of the number of residential land transactions between counties |
| Residential land transaction area ($X_{10}$) | $X_{10}$ denotes the total area of residential land transactions in county $i$ in year $j$ | Allows cross-sectional comparisons of the residential land transaction area between counties |

method and Akaike information criterion (AIC) are frequently used. The CV method is applied as follows:

$$CV = \sum_{i=1}^{n} [y_i - \breve{y}_{\neq i}(b)]^2 \qquad (4)$$

where $y^{\wedge}_{\neq i}$ is the fitted value of $y_i$ under bandwidth b and n is the number of observations. Different bandwidth values correspond to different CV values, and the bandwidth corresponding to the minimum CV value is optimal.

The AIC can measure the goodness of fit of the OLS regression model and is defined as follows:

$$AIC = 2n\ln(\check{s}) + n\ln(2p) + n\frac{n + tr(s)}{n - 2 - r(s)} \tag{5}$$

where σ^ is the maximum likelihood estimate of the variance in the random error term. $tr(s)$ is the bandwidth function, and $S$ is the trace of the hat matrix. The AICc value is obtained by correcting the calculated AIC value, and the bandwidth with a lower AIC or AICc value is optimal. With the bandwidth thus determined, the GWR model can better fit the data.

## Results and discussion

### Analysis of spatial and temporal characteristics of residential land prices

**ESDA and kriging interpolation analysis of residential land prices.** Through the spatial statistical analysis of the residential land price data, we can evaluate the characteristics of the data distribution and, when necessary, use logarithmic transformation to obtain normally distributed data that satisfy the assumption of variability. This paper uses the Geostatistical Analyst tool in ArcGIS 10.2 to perform spatial statistical analysis and examine whether the sample of residential land prices in each county conforms to the normal distribution through histograms and QQ-plot normal distribution graphs. In this paper, the characteristics of the initial and transformed land prices of 174 counties in Beijing-Tianjin-Hebei are described, with the results shown in **Table 3** and Fig 2.

Table 3 shows that the residential land price skewness values of the counties in Beijing-Tianjin-Hebei for 2014 to 2017 are in the range of 2.93~3.68 and the kurtosis values in the range of 10.94~16.83. The initial land price spatial structure does not follow a normal distribution. After logarithmic transformation, the skewness value of the land price is distributed between 1.29 and 1.47 and the kurtosis value between 4.07 and 5.12. After the transformation, the skewness and kurtosis gaps are significantly smaller than the initial values, the sample data are more symmetric, and the steepness of the distribution is reduced. Fig 2 shows that the distribution of the transformed residential land price data is close to the trend line of a normal distribution. Although there are some individual deviations, overall, the data approximate a normal distribution. In short, the transformed sample data show strong normal distribution characteristics, enhancing the reliability of the kriging spatial difference analysis. The interpolation results of the residential land price sample under the spherical model for each county in Beijing, Tianjin, and Hebei are as follows (Fig 3).

**Table 3. Statistics of sample characteristics of residential land prices in Beijing-Tianjin-Hebei region.**

| time | Before and after transformation | maximum value | minimum value | average value | standard deviation | skewness | kurtosis |
|------|-------------------------------|---------------|---------------|---------------|-------------------|----------|----------|
| 2014 | initial value | 28504.00 | 354.00 | 2750.20 | 4429.50 | 3.68 | 16.83 |
|      | Log transformation | 4.45 | 2.55 | 3.22 | 0.37 | 1.38 | 5.12 |
| 2015 | initial value | 34718.00 | 408.00 | 3148.20 | 5182.70 | 3.57 | 16.54 |
|      | Log transformation | 4.54 | 2.61 | 3.32 | 0.38 | 1.47 | 4.96 |
| 2016 | initial value | 37565.00 | 531.00 | 3889.20 | 6476.70 | 3.29 | 14.02 |
|      | Log transformation | 4.57 | 2.72 | 3.33 | 0.40 | 1.41 | 4.47 |
| 2017 | initial value | 40321.00 | 554.00 | 4989.70 | 8009.20 | 2.93 | 10.94 |
|      | Log transformation | 4.61 | 2.74 | 3.43 | 0.41 | 1.29 | 4.07 |

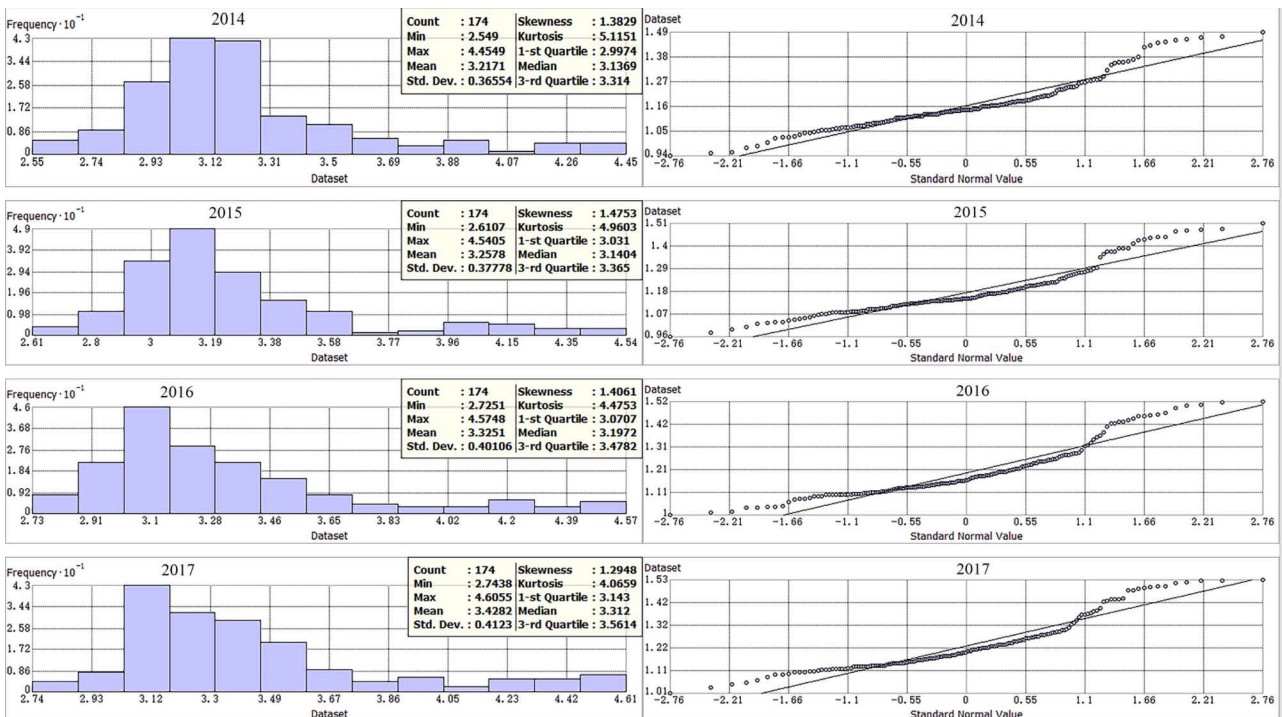

**Fig 2. The histogram of residential land price samples and QQ-plot distribution map from 2014 to 2017.**

In *Fig 3*, the spatial distribution characteristics of residential land prices in the Beijing-Tianjin-Hebei region and their changing trends are relatively obvious. First, residential land prices in Beijing and Tianjin are significantly higher than those in other regions, and the difference between areas with high and low land prices tends to increase over time. In 2014, the residential land price in Beijing-Tianjin and surrounding areas ranged from 10,789 to 27,910 yuan/m², and that in the western mountain counties ranged from 356 to 748 yuan/m². In 2017, residential land prices in Beijing-Tianjin and surrounding areas ranged from 18,827 to 40,523 yuan/m², while land prices in low-value areas ranged from only 685 to 936 yuan/m². During this period, attracted by potentially higher wages and prices, labor, capital and other productive factors from the counties surrounding Beijing and Tianjin clustered in these two cities, thus generating agglomeration benefits that enhanced the external competitiveness of these poles of economic development. In turn, the economic strength and residential land prices of Beijing, Tianjin and the surrounding counties outperformed those of other areas and grew rapidly. Beijing-Tianjin-Hebei created a coordinated development plan, with the Zhangjiakou and Chengde area and the Yanshan-Taihang Mountains in Northwest Hebei designated as ecological conservation areas with a focus on enhancing ecological security, water conservation, tourism and leisure functions. The functional positioning of the region has kept residential land prices low and slow to rise in some counties. Second, a development trend of residential land price polycentricity gradually emerged, With the gradual increase in locational relevance. An overview of the residential land price equivalent surface in each year reveals that except for two land price peaks in Beijing and Tianjin, subcenters have gradually formed mainly in urban areas in Hebei Province. In 2014, a core of six subcenters—the Shijiazhuang, Tangshan, Baoding, Handan, Zhangjiakou, and Hengshui urban areas—was initially formed. Land prices in the subcentral areas are between 2,381 and 4,610 yuan/m².

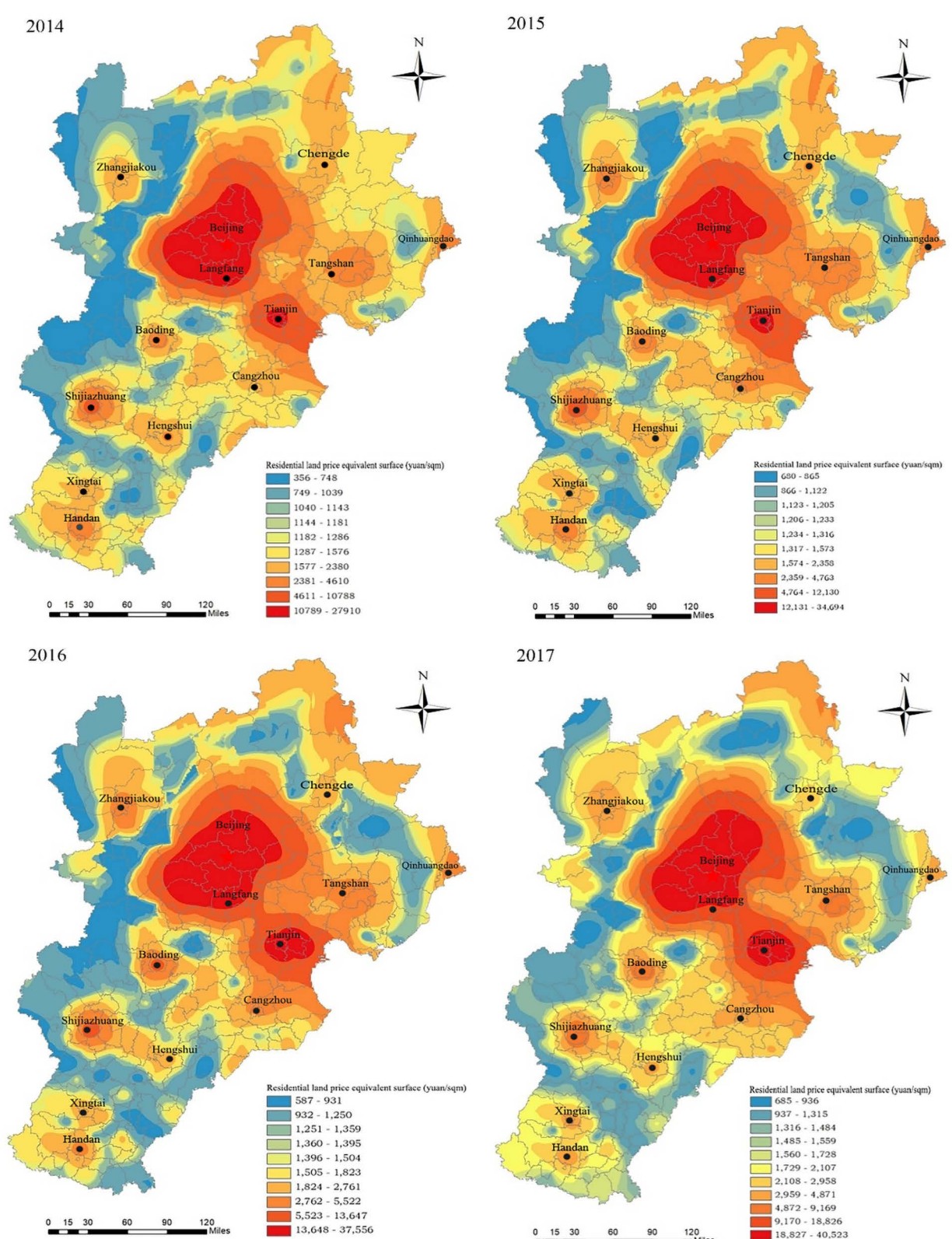

**Fig 3. Equivalent map of residential land prices in the Beijing-Tianjin-Hebei region from 2014 to 2017.**

Table 4. Statistical table of spatial autocorrelation analysis of residential land prices in Beijing-Tianjin-Hebei from 2014 to 2017.

| Various indicators | 2014 | 2015 | 2016 | 2017 |
|---|---|---|---|---|
| Moran's I | 0.4346 | 0.4762 | 0.5131 | 0.5421 |
| expected index | -0.0058 | -0.0058 | -0.0058 | -0.0058 |
| variance | 0.0010 | 0.0010 | 0.0010 | 0.0010 |
| z-value | 14.1771 | 15.5008 | 16.5558 | 17.3146 |
| p-value | 0.0000 | 0.0000 | 0.0000 | 0.0000 |

In 2017, Langfang did not form its own central point because of its integration with Beijing. Except for Chengde and Qinhuangdao, eight subcenters formed around their urban cores in Hebei province; among these subcenters, Shijiazhuang, Baoding, and Tangshan had relatively high land prices ranging from 4,872 to 9,169 yuan/m$^2$ in the central city and 2,959 to 4,871 yuan/m$^2$ in other urban centers. The change in subcenter land prices has led to overall improvements in land prices in the surrounding counties, and the gradient of land price in many cities has started to overlap and become linked, with a "point-axis" land price development pattern gradually forming and the location correlation increasing. There are six such point-axis residential land price development patterns, including Beijing-Tianjin-Langfang, Beijing-Tangshan, Beijing-Baoding, Tangshan-Tianjin-Cangzhou, Shijiazhuang-Hengshui, and Xingtai-Handan.

**Global spatial autocorrelation analysis of residential land prices.** This paper uses the global Moran index to analyze the global spatial autocorrelation of residential land prices in Beijing-Tianjin-Hebei from 2014 to 2017; the results are shown in Table 4.

As seen from Table 4, the Moran's I distribution of the residential land price data points for each year ranges from 0.4346 to 0.5421, with all values greater than 0. With a 99% confidence interval and a two-sided test threshold of 2.58, the Z-values are distributed from 14.1771 to 17.3146, far exceeding the threshold, with p-values less than 0.01 (significant). This result shows that the residential land prices of each county in Beijing, Tianjin, and Hebei have strong spatial clustering characteristics and global autocorrelation and are positively spatially correlated, which lays the foundation for the construction of the GWR model.

## Analysis of the driving effect of land market factors on residential land price

**Determination of the optimal bandwidth of the GWR model.** The GWR model provides two spatial weight function calculation methods in the spatial weight function calculation module, namely, the fixed kernel type (FIXED) and the adaptive kernel type (ADAPTIVE). Different methods of calculating the spatial weight function make the GWR model fit differently, with the more appropriate fit determined by the analysis and testing of the model parameters. In this paper, both kernel types are used, and the CV method and AIC are applied to compare and analyze the optimal bandwidth. The residential land price is the dependent variable, and the bandwidth is calculated with 10 residential land market indicators. The results are as follows (Table 5).

**The contribution of land market factors to residential land price changes.** With the residential land prices in our sample of 174 counties and districts in Beijing-Tianjin-Hebei as dependent variables and the 10 residential land market indicators as explanatory variables, 174×10 regression coefficients can be obtained by running the GWR model. The results are shown in Table 6. The maximum, minimum, mean, upper quartile, median, and lower quartile of the regression coefficients and the local R$^2$ and standard residuals reveal the magnitude and dispersion of the contribution of each explanatory variable to the spatial differentiation

**Table 5. GWR model bandwidth calculation results.**

| Model parameter | Fixed kernel | | Adaptive kernel | |
|---|---|---|---|---|
| | CV | AIC | CV | AIC |
| Bandwidth /Neighbors | 95420.63 | 99083.73 | 127 | 77 |
| ResidualSquares | 13.53 | 14.76 | 42.33 | 26.90 |
| EffectiveNumber | 84.24 | 80.76 | 37.18 | 64.63 |
| Sigma | 0.39 | 0.40 | 0.56 | 0.48 |
| AICc | 270.51 | 270.13 | 321.65 | 306.98 |
| $R^2$ | 0.9218 | 0.9147 | 0.7553 | 0.8561 |
| Adjusted $R^2$ | 0.8493 | 0.8416 | 0.6906 | 0.7723 |

In the comparison of the calculation module results, the residual square and AICc values for the fixed kernel type are relatively small, and the $R^2$ and adjusted $R^2$ are relatively large, indicating that the fit of the fixed kernel is better than that of the adaptive kernel calculation module. The comparison of calculation methods shows CV method's slightly better residual square, AICs, $R^2$, and adjusted $R^2$ parameters. Therefore, the GWR model in this paper adopts the fixed kernel type module and the CV method and obtains an optimal bandwidth of 95,420.63 m. At this time this model offers the best available smoothing degree.

of residential land price levels and the effect of the fit across the variables. The Monte Carlo method was also used to test the significance of each influence factor for spatial non-stationarity and to estimate the p-values of the regression coefficients. Among the regression parameters, LocalR2 characterizes the goodness of fit of the local regression model to the observed values. StdResid is the standard residual, which is the difference between the observed and predicted values (fitted value), that is, the difference between the actual observed value and the regression estimate, with a better fitting effect when the value distribution is within (-3, 3).

Table 6 shows that the mean LocalR2 is 0.7067, and the median is 0.7262. The standard residuals are distributed in (-2.6421, 2.5056), the mean is -0.0221, the median is 0.0255, and the absolute values are mostly less than 3. The data show that the fit of the GWR model is satisfactory and that the residential land market indicators have a strong ability to explain residential land prices. Among the indicators, the regression coefficients of $X_9$ and $X_{10}$ are

**Table 6. GWR model regression coefficient statistics.**

| explanatory variables | maximum value | minimum value | mean value | standard deviation | Upper quartile | Median | Lower quartile | p-value |
|---|---|---|---|---|---|---|---|---|
| $X_1$* | 0.4390 | -0.2448 | 0.0447 | 0.1156 | -0.0090 | 0.0201 | 0.0864 | 0.018 |
| $X_2$** | 0.2443 | -0.4881 | -0.0307 | 0.1127 | -0.0854 | -0.0391 | 0.0344 | 0.006 |
| $X_3$* | 0.1658 | -0.4325 | -0.1473 | 0.1347 | -0.2345 | -0.1183 | -0.0301 | 0.023 |
| $X_4$* | 0.7627 | -0.6695 | 0.1636 | 0.2301 | 0.0130 | 0.0578 | 0.3391 | 0.002 |
| $X_5$*** | 0.5335 | -0.9289 | -0.0816 | 0.1876 | -0.1294 | -0.0288 | 0.0104 | 0.000 |
| $X_6$*** | -0.0174 | -0.5785 | -0.2315 | 0.1346 | -0.3391 | -0.1817 | -0.1276 | 0.001 |
| $X_7$* | 0.0970 | -0.3183 | -0.0255 | 0.0612 | -0.0326 | -0.0076 | 0.0071 | 0.024 |
| $X_8$*** | 0.6855 | -0.0174 | 0.2456 | 0.1913 | 0.0772 | 0.1940 | 0.4294 | 0.000 |
| $X_9$*** | -0.0449 | -2.9304 | -0.4311 | 0.5633 | -0.4589 | -0.2651 | -0.0775 | 0.000 |
| $X_{10}$*** | 2.5790 | 0.1767 | 0.6386 | 0.4727 | 0.2633 | 0.5654 | 0.7981 | 0.000 |
| LocalR² | 0.9031 | 0.4780 | 0.7067 | 0.1061 | 0.6425 | 0.7262 | 0.7927 | / |
| StdResid | 2.5056 | -2.6421 | -0.0221 | 1.0834 | -0.3746 | -0.0255 | 0.2979 | / |

Note

*** represents the 0.1% significance level

** represents the 1% wetness level

* represents the 5% significance level.

above 0.43 on average, with the largest contribution; these variables have the strongest explanatory power in relation to changes in the residential land price level. The mean values of the regression coefficients of $X_3$, $X_4$, $X_6$, and $X_8$ range from 0.14 to 0.25, again representing large contributions to the residential land price level. The mean values of the regression coefficients of $X_1$, $X_2$, $X_5$, and $X_7$ are below 0.09, representing the weakest effects on the residential land price level. The p-value of each influence factor of the Monte Carlo test is less than 0.05, so each influence factor has significant spatial non-stability, where X5, X6, X8, X9, and X10 have the highest significant level, X2 is the second, and X1, X3, X4, and X7 have the weakest significant level.

**The driving mechanism of land market factors on residential land prices.** To visually show the spatial characteristics of the correlation between the residential land market indicators and residential land prices, we use ArcGIS software to generate regression coefficient distribution maps (Figs 4, 6 and 8). The indicators of the residential land market are divided into three types based on coefficient size and the different spatial distribution characteristics.

1. Positive factors and mechanism

Fig 4 shows that the regression coefficients of the utilization rate of stock land (X4), revenue from residential land transfers (X8), and the residential land transaction area (X10) range from -0.6695 to 0.7627, -0.0174 to 0.6855, and 0.1767 to 2.5790. The coefficients of the X4 and X8 indicators are positive for most counties, and those of the X10 indicators are positive for the entire region. This indicates that the three indicators are significantly positively correlated with residential land prices.

The revenue from land transfers and the residential land transaction area are important indicators of local government's dependence on "land finance". In recent years, the land finance model has been an essential tool whereby local governments in China accumulate urban capital, promote urbanization and economic development, and generate wealth from the land. Under this model, land revenue and increases in transaction areas can meet the financial needs of local governments seeking to build infrastructure and public facilities, boost infrastructure investment and construction, promote local economic development, and improve the urban living environment and supporting facilities to drive asset appreciation, thus driving up land prices. In this process, local governments obtain more revenue from land concessions, while the expectation of land price increases drives up land demand in the market, leading to an expansion of land supply, dependence on land finance, and land price increases and thus forming a cycle. This dynamic is particularly obvious in Beijing-Tianjin and its surrounding areas, which are characterized by rapid social and economic development, dense populations, and convenient intercity linkages.

In the context of the new urbanization strategy and the "new normal" economic development approach, excavation of the stock of land is inevitable. In 2014, the former Ministry of Land and Resources put forward strict requirements designed to increase and revitalize this stock and optimize the structure and improve the efficiency of the land market, aiming to vigorously promote the conservation and intensive use of land and accelerate the transformation of land use and economic development methods. Under the combined effects of the market supply and demand mechanism and macro-management, on the one hand, On the one hand, the revitalization and utilization of the stock of construction land improves the development intensity of urban construction land, realizes the optimal allocation of production factors and land resources, and optimization of the land structure within the town and the improvement of the comprehensive land carrying capacity contribute to the increase of land prices. On the other hand, utilization of the stock of land is subject to strict controls on the total amount of construction land and reductions in the scale of new construction land. High demand for land

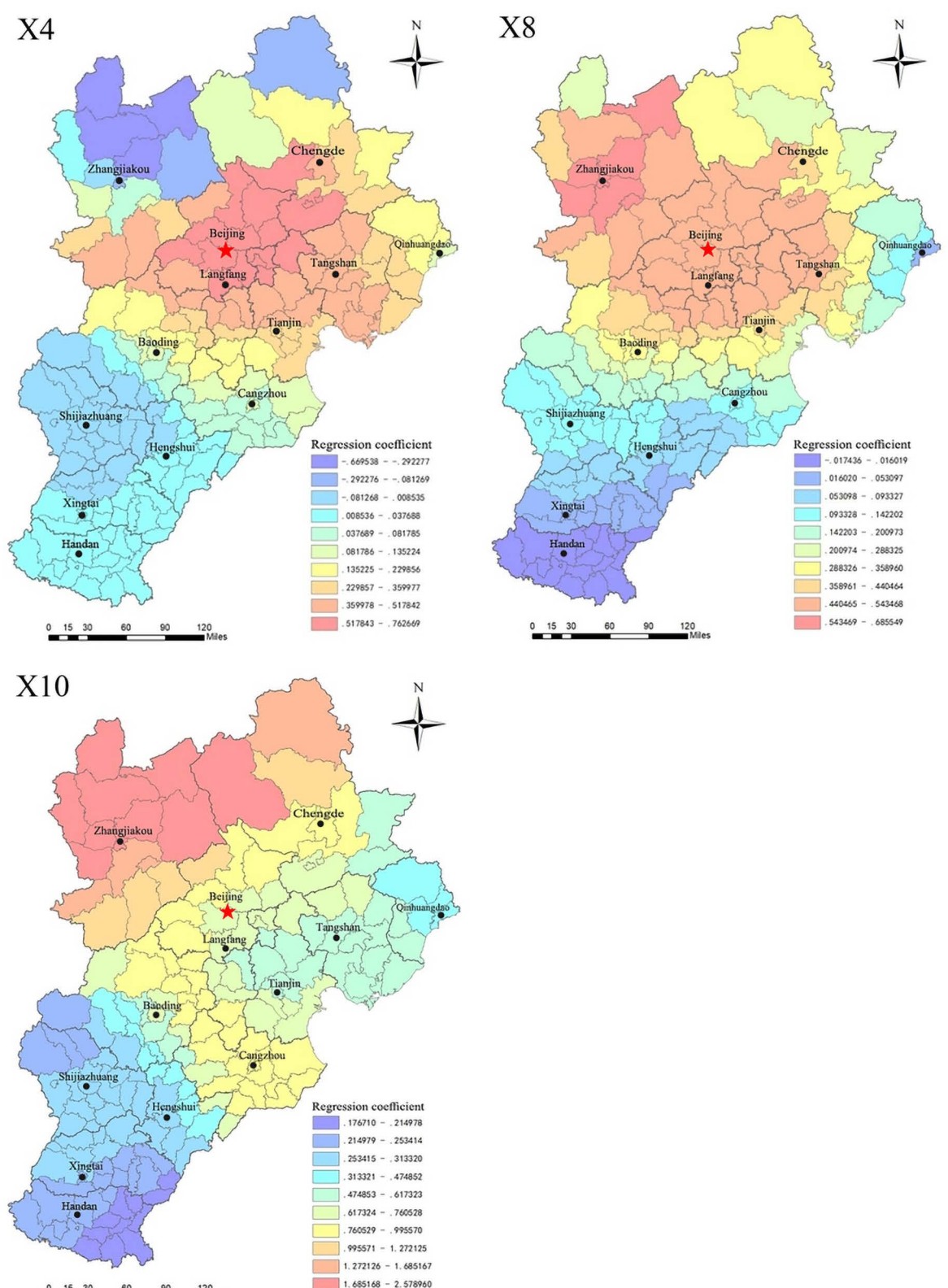

**Fig 4. Regression diagram of market indicators for the significant positive driving of residential land prices.**

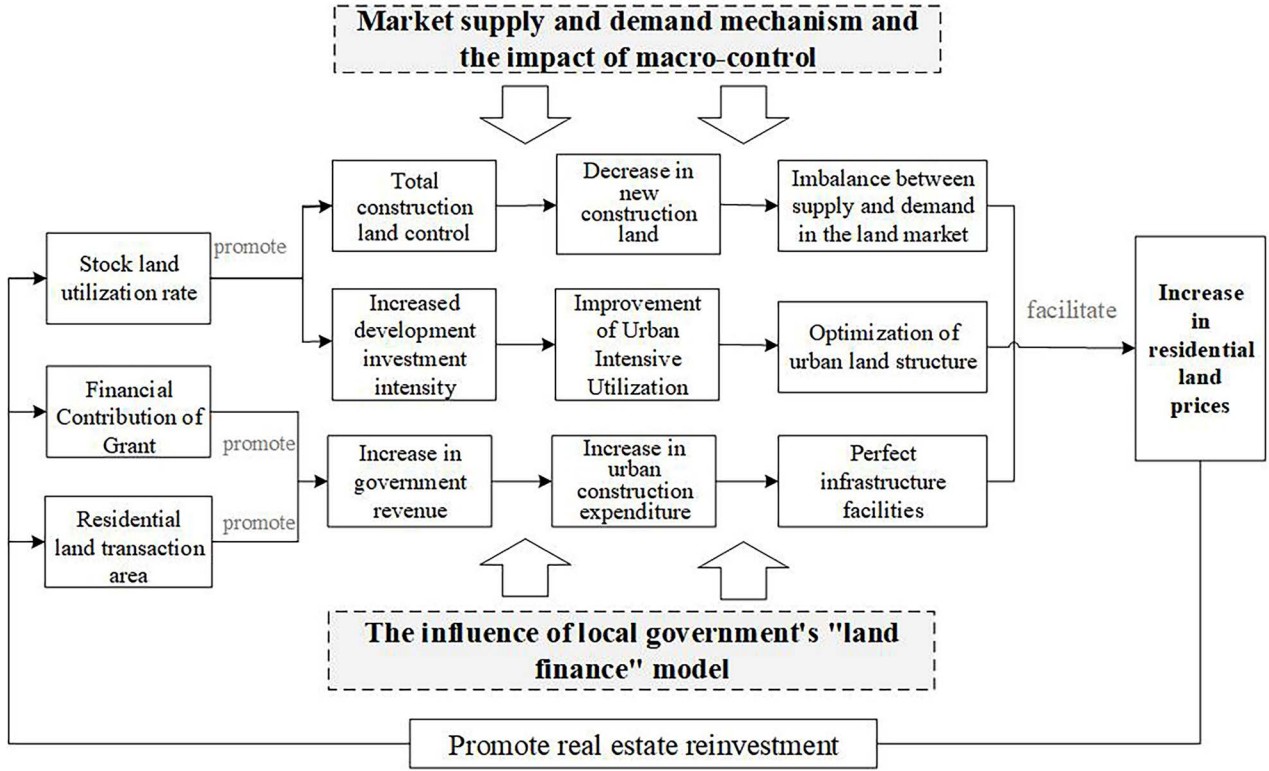

**Fig 5. The mechanism of the factors promoting the rise of residential land prices.**

and tightened land supply under rapid urbanization development impact supply and demand relationship, causing land prices to rise.

The mechanism whereby the $X_4$, $X_8$ and $X_{10}$ indicators affect residential land prices is as follows (Fig 5).

2. Negative factors and mechanism

Fig 6 shows that the coefficients on the degree of marketization of the residential land supply ($X_3$), residential land supply rate ($X_5$), per capita residential land supply area ($X_6$), and residential land transaction frequency ($X_9$) are in the ranges of -0.4325~0.1658, -0.9289~0.5335, -0.5785~-0.0173, and -2.9304~-0.0449. The coefficients of $X_3$ and $X_5$ are negative for most counties, and those of $X_6$ and $X_9$ are negative for all counties, indicating a clear negative correlation between the four indicators and residential land prices; therefore, improvements to these factors can suppress residential land prices.

The effect of the land supply rate and per capita residential land supply area on residential land prices may be related to the market supply and demand mechanism. Local governments determine the supply plan of residential land based on housing construction and annual implementation plans, combined with the total cumulative saleable area of housing, the total amount of residential land not yet being used for construction, and other indicators. The residential land supply plan represents the amount needed to fulfill demand for land in each region, while the effective land supply rate reflects the actual supply for development and investment by real estate enterprises, the level of which reflects not only the efficiency of land resource allocation but also the supply and demand situation in the land market. An increase in the land supply rate

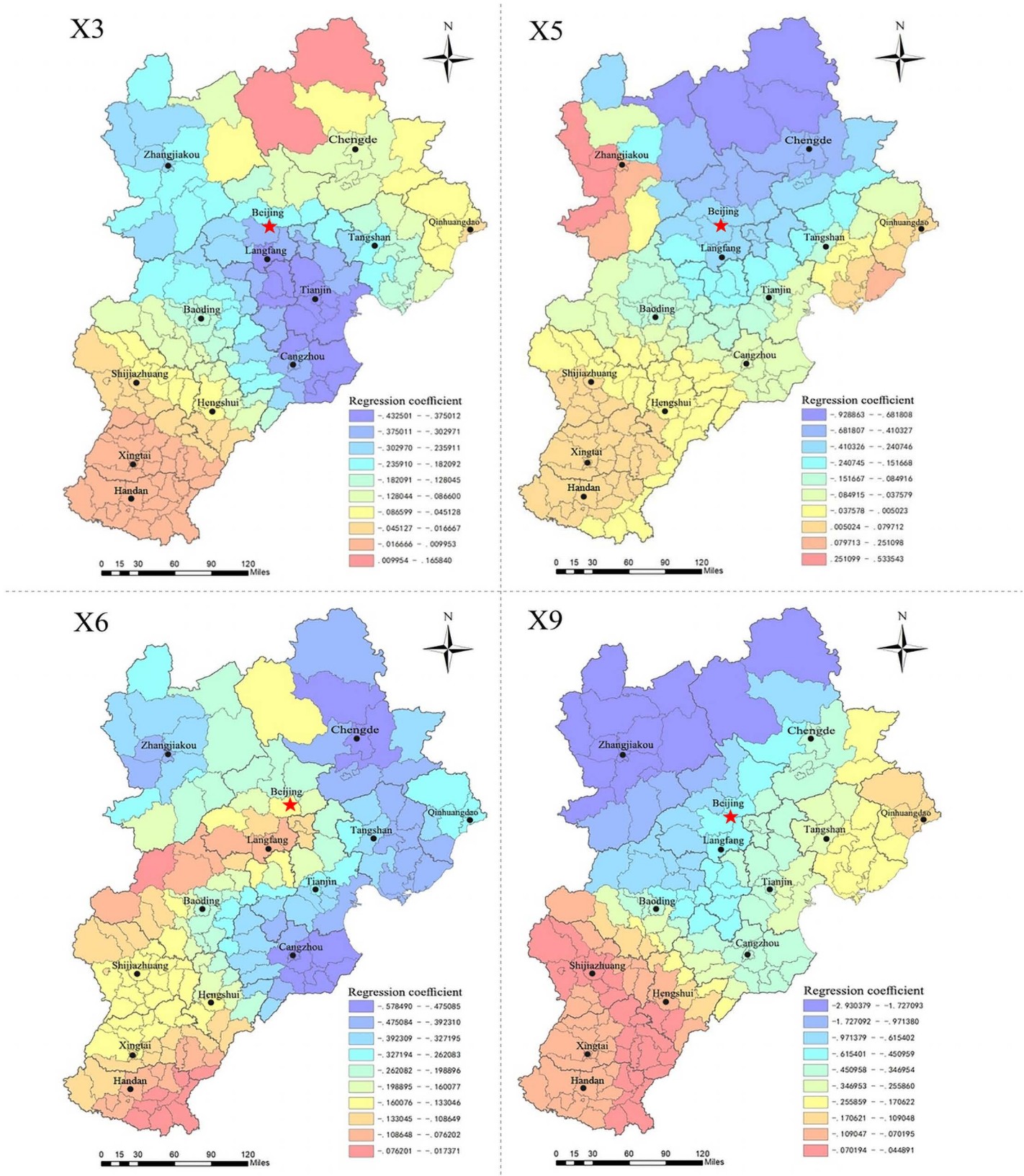

**Fig 6. Regression diagram of significantly negative driving market indicators for residential land prices.**

can fulfill the strong demand for real estate land, alleviate supply shortfalls in the residential land market, and thus play a role in curbing or stabilizing land prices. The residential land supply area per capita reflects the relationship between the supply of residential land in each region and potential land demand among the population. A continuous increase in the supply of residential land per capita can bring about market imbalances, leading to an excess residential land and housing inventory, which in turn inhibits real estate prices.

The degree of marketization and transaction frequency's restraining effect on land prices may be related to the continuous improvement and strengthening of residential land supply management measures in recent years. Under the Chinese land market system, local governments have a monopoly over the development and operation of the primary land market, making them "rule makers" in land acquisition and supply management. As the real estate market has become more regulated in recent years, the control of the land market has become an important tool for the government to stabilize land and housing prices. For example, in 2014, the former Beijing Municipal Bureau of Land and Resources proposed a series of land transfer improvement measures, adopting more comprehensive bid evaluation methods based on "best comprehensive conditions", in which the primary bid evaluation conditions considered are factors such as the bid price; payment schedule; development and construction cycle; degree of land savings and use intensification; enterprise qualifications, performance, and financial status; share of policy-based housing in a construction project; and commitments to control future commercial housing sales prices. Second, to improve preapplication screening, a reasonable price range for land is set and the reserve price determined based on the comprehensive evaluation of regional land prices by social intermediaries, taking into account expert input and collective decision-making in the relevant departments. Third, the importance of the bid price factor is further deemphasized, such that the final designation of the land transferee now follows the principle of best comprehensive conditions. It has been proven that this comprehensive bid evaluation method can guide enterprises to participate in land bidding rationally, prevent speculative behavior and unreasonable premiums, and help stabilize market expectations and land prices. Similar to increases in marketization, an increase in the number of transactions facilitates the government's flexibility in determining plot sizes and layouts and controlling the scale of residential land transactions and the pace of supply; this, in turn, helps stabilize and balance the supply of residential land, prevent spikes in land prices and stabilize market expectations. A higher degree of marketization not only reflects a higher proportion of bid, auction, and listing transfers but also serves as a more complete land supply control measure. A higher transaction frequency reflects the flexibility and controllability of the residential land supply. Thus, the local government's monopoly in the primary land market and the continuous improvement and strengthening of residential land supply management measures may be the intrinsic reasons for the rapid rise in land prices curbed by the two indicators.

The mechanism whereby $X_3$, $X_5$, $X_6$, and $X_9$ drive residential land prices can be summarized as follows (Fig 7).

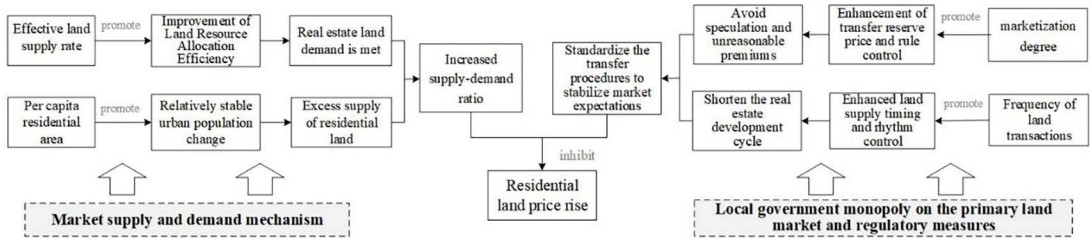

**Fig 7. The mechanism of the inhibitory factors of residential land price rise.**

3. Spatial characteristics of the driving effect

Regardless of whether the effect of the land market factors considered is positive or negative, their correlation with residential land prices has similar characteristics across space. The correlation across the central and northern regions of Beijing-Tianjin-Hebei (especially the surrounding areas of Beijing-Tianjin) is relatively significant, while in the southern region, the correlation is weak. On the one hand, Beijing and Tianjin are characterized by rapid social and economic development, dense populations, and more convenient intercity linkages. The areas around Beijing and Tianjin have great development potential because they can relieve Beijing's noncapital functions and facilitate industrial upgrading and transformation, and their residential land market is also more active. According to the basic information, the total area of residential land granted in Beijing, Tianjin, and six surrounding areas (Langfang, Tangshan, Baoding, Cangzhou, Zhangjiakou, and Chengde) accounted for 71% of the residential land area of the entire Beijing-Tianjin-Hebei region during 2014–2017. The relatively active land market and adequate land revenues in the northern region have greatly contributed to the development of its real estate industry, which makes land market factors more sensitive to the driving effect of residential land prices. On the other hand, Beijing and its neighboring cities, as a key area targeted by land market regulations, have more stringently implemented land plans, use of stock land, and standardized land market behavior. For example, the overall utilization rate of stock residential land in Beijing reached 41% during 2014–2017, significantly higher than the level in other regions. In short, the strict macro-management policies in the Beijing-Tianjin area make the effect of land market factors on residential land prices more obvious.

4. Factors with weaker correlations

The regression coefficients of the land price–GDP elasticity coefficient ($X_1$), residential land supply ratio ($X_2$), and proportion of land for affordable housing ($X_7$) are in the ranges of -0.2448~0.4390, -0.4881~0.2443, and -0.3183~0.0970. Spatially, as shown in **Fig 8**, the 3 indicators do not show universal characteristics with residential land prices in the region. The positive correlations between $X_1$ and $X_2$ and residential land prices are mainly found in Chengde, Tangshan, and Qinhuangdao, while negative correlations are evident in Zhangjiakou, Baoding, and some counties in Beijing. Most of the areas with a positive correlation between the $X_3$ index and residential land prices are in the southwestern part of the Beijing-Tianjin-Hebei region (Shijiazhuang, Hengshui, Xingtai, and the western part of Baoding, etc.), but the positive correlation is most significant in Chengde County and Pingquan County of Chengde City; the negative correlation appears in other areas of Beijing-Tianjin-Hebei. The spatial distribution of the regression coefficients shows that these three indicators are not strongly correlated with residential land prices and do not have an obvious effect.

## Conclusions

This study aims to study the spatial and temporal evolution of residential land price levels in the Beijing-Tianjin-Hebei urban agglomeration and the driving effect of land market factors on residential land prices, so as to point out the direction for future synergistic land market development in the region. We used residential land prices and land market data of 174 counties and districts in Beijing, Tianjin, and Hebei from 2014–2017, and we adopted exploratory spatial data analysis methods and kriging interpolation analysis to derive the Spatio-temporal variation characteristics of residential land prices in the Beijing-Tianjin-Hebei region. The influencing factors that are highly correlated with residential land prices are selected around four attributes: land market price mechanism, supply and demand, land supply structure, and

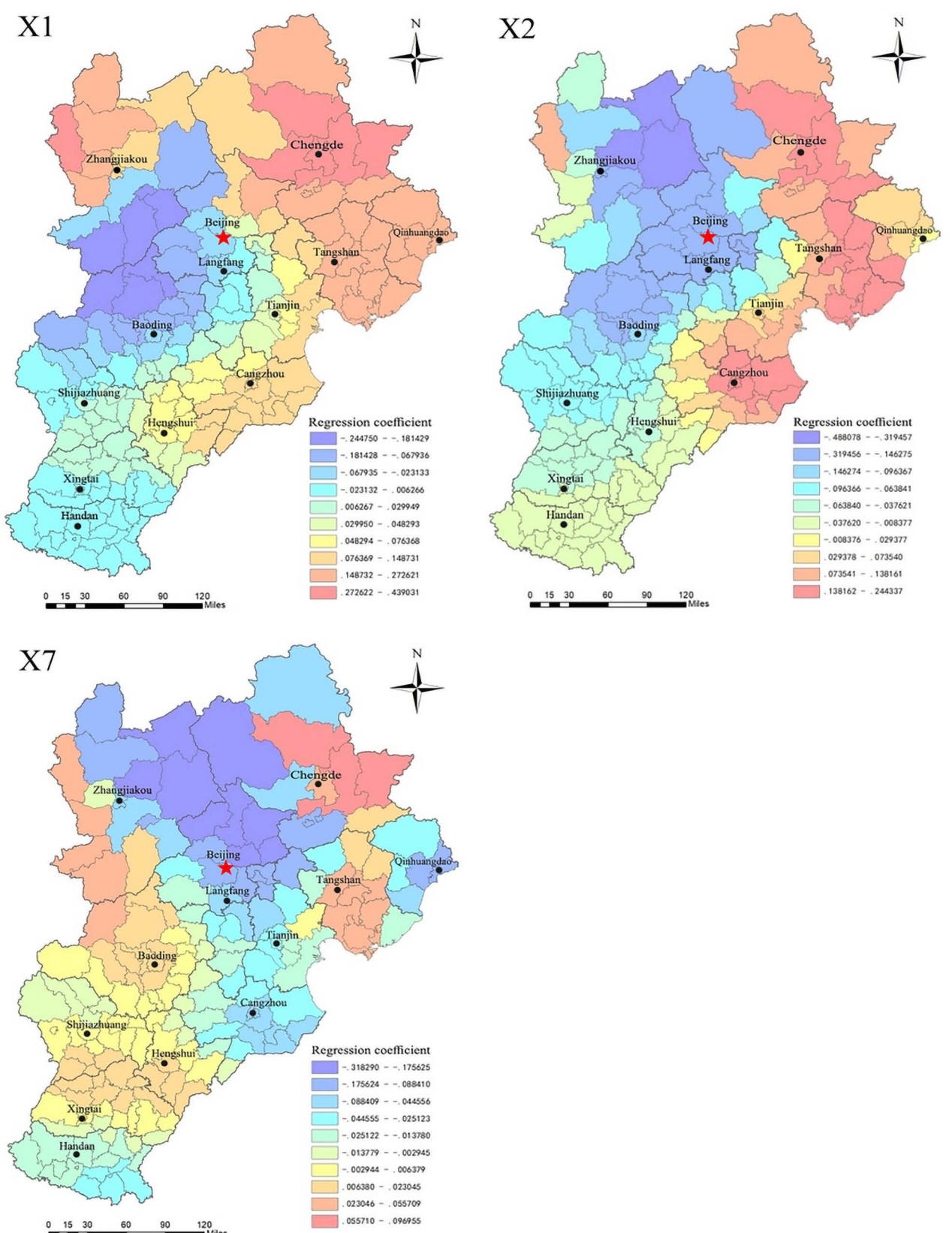

**Fig 8. Regression diagram of indicators with weak correlation to residential land prices.**

market activity, and the driving mechanism of land market factors on residential land prices is revealed with the help of GWR model. The analysis shows that (1) residential land prices in Beijing and Tianjin are significantly higher than those in other regions and that the spatial differences between the high and low land prices tend to increase over time. Over time, a development trend of residential land price polycentricity gradually emerged, and the locational correlation has gradually increased. In addition to the two land price peaks in Beijing and Tianjin, subcenters have gradually formed—mainly in urban areas in Hebei Province. With the advance of cooperative development across Beijing-Tianjin-Hebei, six point-axis residential land price development patterns have formed: Beijing-Tianjin-Langfang, Beijing-Tangshan, Beijing-Baoding, Tangshan-Tianjin-Cangzhou, Shijiazhuang-Hengshui, and Xingtai-Handan. (2) Under the land finance operational model of local governments in China, the revenue contribution of land transfers and the increase in transaction areas have become the main driving force behind the rise in residential land prices, and the government's reliance on "land finance" and rising residential land prices form a path cycle. The government's control of the total supply of new residential land through regulatory policies and development of the stock of land have also contributed to the rise in residential land prices. (3) Under the supply and demand mechanism in the land market, increases in the effective land supply rate and per capita residential land supply area can dampen the rise of residential land prices. At the same time, with the deepening of marketization in the land market in China in recent years, land transactions have become more rational and standardized, which has also helped stabilize the rise in residential land prices.

Based on the above research, this article offers the following policy recommendations for the effective regulation of residential land prices. (1) Government departments should support the construction of subcenter cities and promote the formation of a polycentric spatial structure in Beijing, Tianjin, and Hebei. On the one hand, the infrastructures of subcenter cities should be improved, high-standard schools and hospitals and sufficient jobs provided, and intercity rapid rail transportation developed between subcenters and main centers. On the other hand, policymakers should pay attention to land price differences within cities and use the land market price discovery mechanism to guide the development of industrial clustering in different districts within cities to rationalize the spatial differentiation of land prices in the Beijing-Tianjin-Hebei region. (2) Policymakers should deepen the reform of the fiscal and taxation system, accelerate the process of real estate tax collection, and gradually shift local governments' dependence on land finance toward a reformed real estate tax. This can alleviate the financial pressure on local governments and reduce their dependence on land transfer income, thereby stabilizing residential land prices. (3) Policymakers should further improve the land supply management system, e.g., by developing a long-term, stable, and transparent land supply plan and increase the effective land supply rate to reasonably guide market expectations. At the same time, it is important to standardize the market orientation of land transactions and continuously explore a land management mechanism that combines market behavior and macroregulation and management to promote the stable and healthy development of the housing market. (4) According to the current market control background of "classification guidance and local policy", different land control policies should be formulated and implemented according to the current situation of supply and demand in the land market in different types of cities. Examples include the following: ① Land supply plans should be formulated and implemented according to the local situation, and the change in land supply should reflect the population size and per capita income, thus guaranteeing a balanced supply and demand in the land market. ② Postsupply supervision of land supply should be strengthened to avoid land idleness, optimize land use structure, excavate the stock of land within the city, and improve land intensification and economization. ③ Optimization of the land market

supply method should be continued and the land transfer method should be selected in a scientifically informed manner according to the use purpose of the land transferred to achieve fairness, justice, and openness as much as possible, promote the structural reform of the land supply side, strengthen the proportion of the supply of guaranteed and policy housing, and realize the benign development of the residential market.

## Supporting information

**S1 Fig. Data of the sample histogram and QQ-plot distribution map of residential land prices in 2014–2017.** The normal distribution of residential land prices in 2014–2017.
(ZIP)

**S2 Fig. Data of the residential land price equivalence map 2014–2017.** Kriging interpolation results from 2014–2017.
(ZIP)

**S3 Fig. Data of regression diagram of market indicators for the significant positive driving of residential land prices.** Distribution map of the driving effect of X4, X8, X10 on residential land prices.
(ZIP)

**S4 Fig. Data of regression diagram of significantly negative driving market indicators for residential land prices.** Distribution map of the driving effect of X3, X5, X6, X9 on residential land prices.
(ZIP)

**S5 Fig. Data of regression diagram of indicators with weak correlation to residential land prices.** Distribution map of the driving effect of X1, X2, X7 on residential land prices.
(ZIP)

**S1 Table. Statistical data on the characteristics of residential land prices in the Beijing-Tianjin-Hebei region.** The normal distribution of residential land prices in 2014–2017.
(XLS)

**S2 Table. Data from the statistical table of spatial autocorrelation analysis of residential land prices in Beijing-Tianjin-Hebei from 2014–2017.** Global Moran Index Results 2014–2017.
(XLS)

**S3 Table. GWR model regression coefficient data.** 10 driving factors GWR regression results.
(XLSX)

## Acknowledgments

We greatly appreciated Prof. Zhongjiang Feng, Lecturer Man Zhang, and A.P.Yanqing Liang for their valuable comments at various stages of this study.

## Author contributions

**Conceptualization:** Xiaomiao Feng.

**Funding acquisition:** Can Li, Jingfeng Ge.

**Project administration:** Can Li.

**Software:** Pengfei An.

**Writing – original draft:** Pengfei An.

**Writing – review & editing:** Pengfei An, Yajing Duan, Jingfeng Ge, Xiaomiao Feng.

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
