## [Decision Letter · Decision Letter 0]

15 Jul 2021

PONE-D-21-20597

Inter-metropolitan land price characteristics and pattern in the Beijing-Tianjin-Hebei urban agglomeration, China

PLOS ONE

Dear Dr. an,

Thank you for submitting your manuscript to PLOS ONE. After careful consideration, we feel that it has merit but does not fully meet PLOS ONE’s publication criteria as it currently stands. Therefore, we invite you to submit a revised version of the manuscript that addresses the points raised during the review process.

We look forward to receiving your revised manuscript.

Kind regards,

Jun Yang

Academic Editor

PLOS ONE

Journal Requirements:

[NO].

3. Please ensure that you refer to Figure 1, 7 and 9 in your text as, if accepted, production will need this reference to link the reader to the figure.

4. We note that Figures 1, 5, 6, 8 and 10 in your submission contain map images which may be copyrighted. All PLOS content is published under the Creative Commons Attribution License (CC BY 4.0), which means that the manuscript, images, and Supporting Information files will be freely available online, and any third party is permitted to access, download, copy, distribute, and use these materials in any way, even commercially, with proper attribution. For these reasons, we cannot publish previously copyrighted maps or satellite images created using proprietary data, such as Google software (Google Maps, Street View, and Earth). For more information, see our copyright guidelines: http://journals.plos.org/plosone/s/licenses-and-copyright.

You may seek permission from the original copyright holder of Figures 1, 5, 6, 8 and 10 to publish the content specifically under the CC BY 4.0 license.

If you are unable to obtain permission from the original copyright holder to publish these figures under the CC BY 4.0 license or if the copyright holder’s requirements are incompatible with the CC BY 4.0 license, please either i) remove the figure or ii) supply a replacement figure that complies with the CC BY 4.0 license. Please check copyright information on all replacement figures and update the figure caption with source information. If applicable, please specify in the figure caption text when a figure is similar but not identical to the original image and is therefore for illustrative purposes only.

6. Please upload a copy of Supporting Information Figures and Tables which you refer to in your text on page 33 and 34.

Additional Editor Comments:

Reviewer 1

This study analyzes the spatial and temporal evolution of residential land prices and the factors influencing the land market in the Beijing-Tianjin-Hebei region, using exploratory spatial data analysis (ESDA) and a geographically weighted regression (GWR) model. The argument is sound and the paper is well structured. However, there are still following problems:

1. The keywords are lack.

2. Introduction. Too much. Consider transferring part of it to a separate Literature Review. Literature review needs to be integrated, now it is basically simple lists of existing works. “Lack of a comprehensive analysis of the effects of multiple land market factors on land prices,” "seldom consider the spatial relationship between land prices and land market factors" needs to be based on literature evidence. In fact, there must be some literatures discussing multiple factors and considering spatial relationships. Compared with them, what is your innovation or improvement point?

3. Research area. Per capita GDP and other data should be updated to the latest year.

4. Selection of market indicators for residential land. The selection of indicators needs additional literature support.

5. Methods. Explanations and formulas, such as GWR and AIC, can be reduced appropriately, because they are common methods and do not need detailed introduction. This is not the key part.

6. Results and discussion. Sub-titles. "Analysis of spatial and temporal characteristics of residential land prices" and "Global Spatial Autocorrelation Analysis of Residential Land Prices" are somewhat duplicate, because global spatial autocorrelation belongs to spatial characteristics.

7. The clarity of all images needs to be improved.

8. The language needs polishing by native speakers to make your work better understood.

Reviewer 2

This study aims to investigate the spatial and temporal evolution of residential land price levels in the Beijing-Tianjin-Hebei urban agglomeration and the role of the land market factors in driving residential land prices, so as to provide direction for the future synergistic development of the land market in the region. The MS is innovative in studying the spatially driven mechanism of the land market on residential land price, and the construction of a system of land market indicators affecting residential land prices is scientific and reasonable. It has great potential to show its essential merits for publication, but I recommend the authors to address the following issues:

1. Although the practicality and superiority of the GWR model are described in the introduction section of this paper, the introduction of the GWR model in the research method section is rather messy, and the descriptions of the GWR model in some high-quality papers should be cited to summarize this section to make it concise and clear.

2. This paper is relatively clear in explaining the role and mechanisms of positive and negative drivers of land markets. However, the results of the GWR model lack validation, and the results should be further verified by using, for example, Monte Carlo methods.

3. The conclusion is a good review of the article. However, the policy recommendations need to be specific and should follow the theme of the article, combining the principles of land market indicator system construction, for example, exploring the role of land policy on residential land prices in terms of land market supply and demand, and suggesting corresponding land policy improvements.

4. The introduction section needs to add more up-to-date literature, for example: 1)Wind-sensitive urban planning and design: Precinct ventilation performance and its potential for local warming mitigation in an open midrise gridiron precinct.Journal of Building Engineering 29(1):101145.

2)Contribution of urban ventilation to the thermal environment and urban energy demand: Different climate background perspectives, Science of the Total Environment (2021).

3) Suitability of human settlements in mountainous areas from the perspective of ventilation: a case study of the main urban area of Chongqing, Journal of Cleaner Production (2021).

4)COVID-19: A Comparative Study of Population Aggregation Patterns in the Central Urban Area of Tianjin, China.International Journal of Environmental Research and Public Health 18(4):2135.

5) Understanding land surface temperature impact factors based on local climate zones, Sustainable Cities and Society (2021).

6)Influence of urban morphological characteristics on thermal environment, Sustainable Cities and Society (2021).

5. The language needs to be improved especially the parts of abstract and discussion.

Although the manuscript must be improved, there are still many merits in this manuscript. So, I would like to recommend this manuscript as major revision.

Reviewers' comments:

Reviewer's Responses to Questions

**Comments to the Author**

1. Is the manuscript technically sound, and do the data support the conclusions?

Reviewer #1: Yes

Reviewer #2: Yes

2. Has the statistical analysis been performed appropriately and rigorously?

Reviewer #1: Yes

Reviewer #2: Yes

3. Have the authors made all data underlying the findings in their manuscript fully available?

Reviewer #1: Yes

Reviewer #2: Yes

4. Is the manuscript presented in an intelligible fashion and written in standard English?

Reviewer #1: No

Reviewer #2: Yes

5. Review Comments to the Author

Reviewer #1: This study analyzes the spatial and temporal evolution of residential land prices and the factors influencing the land market in the Beijing-Tianjin-Hebei region, using exploratory spatial data analysis (ESDA) and a geographically weighted regression (GWR) model. The argument is sound and the paper is well structured. However, there are still following problems:

1. The keywords are lack.

2. Introduction. Too much. Consider transferring part of it to a separate Literature Review. Literature review needs to be integrated, now it is basically simple lists of existing works. “Lack of a comprehensive analysis of the effects of multiple land market factors on land prices,” "seldom consider the spatial relationship between land prices and land market factors" needs to be based on literature evidence. In fact, there must be some literatures discussing multiple factors and considering spatial relationships. Compared with them, what is your innovation or improvement point?

3. Research area. Per capita GDP and other data should be updated to the latest year.

4. Selection of market indicators for residential land. The selection of indicators needs additional literature support.

5. Methods. Explanations and formulas, such as GWR and AIC, can be reduced appropriately, because they are common methods and do not need detailed introduction. This is not the key part.

6. Results and discussion. Sub-titles. "Analysis of spatial and temporal characteristics of residential land prices" and "Global Spatial Autocorrelation Analysis of Residential Land Prices" are somewhat duplicate, because global spatial autocorrelation belongs to spatial characteristics.

7. The clarity of all images needs to be improved.

8. The language needs polishing by native speakers to make your work better understood.

Reviewer #2: This study aims to investigate the spatial and temporal evolution of residential land price levels in the Beijing-Tianjin-Hebei urban agglomeration and the role of the land market factors in driving residential land prices, so as to provide direction for the future synergistic development of the land market in the region. The MS is innovative in studying the spatially driven mechanism of the land market on residential land price, and the construction of a system of land market indicators affecting residential land prices is scientific and reasonable. It has great potential to show its essential merits for publication, but I recommend the authors to address the following issues:

1. Although the practicality and superiority of the GWR model are described in the introduction section of this paper, the introduction of the GWR model in the research method section is rather messy, and the descriptions of the GWR model in some high-quality papers should be cited to summarize this section to make it concise and clear.

2. This paper is relatively clear in explaining the role and mechanisms of positive and negative drivers of land markets. However, the results of the GWR model lack validation, and the results should be further verified by using, for example, Monte Carlo methods.

3. The conclusion is a good review of the article. However, the policy recommendations need to be specific and should follow the theme of the article, combining the principles of land market indicator system construction, for example, exploring the role of land policy on residential land prices in terms of land market supply and demand, and suggesting corresponding land policy improvements.

4. The introduction section needs to add more up-to-date literature, for example: 1) Contribution of urban ventilation to the thermal environment and urban energy demand: Different climate background perspectives, Science of the Total Environment (2021). 2) Suitability of human settlements in mountainous areas from the perspective of ventilation: a case study of the main urban area of Chongqing, Journal of Cleaner Production (2021). 3) Understanding land surface temperature impact factors based on local climate zones, Sustainable Cities and Society (2021). 4)Influence of urban morphological characteristics on thermal environment, Sustainable Cities and Society (2021).

5. The language needs to be improved especially the parts of abstract and discussion.

Although the manuscript must be improved, there are still many merits in this manuscript. So, I would like to recommend this manuscript as major revision.

6. PLOS authors have the option to publish the peer review history of their article (what does this mean?). If published, this will include your full peer review and any attached files.

Reviewer #1: No

Reviewer #2: No

---

## [Author Response · Author response to Decision Letter 0]

2 Aug 2021

Editor’s comments:

1.Please ensure that your manuscript meets PLOS ONE's style requirements, including those for file naming.

Response: We appreciate your valuable comment. We have revised the paper in accordance with the requirements of PLOS ONE to make the paper meet the requirements for publication. Comprehensive content of renovation: Font renovation, citation in the text of the image, and naming of supporting information. We believe that our revisions address the reviewers' concerns. We hope our revised manuscript is found acceptable for publication.

2. Thank you for stating the following in your Competing Interests section.

Response: Thank you for your rigorous review. We added the statement "The authors have declared that no competing interests exist" in the cover letter.

3. Please ensure that you refer to Figure 1, 7 and 9 in your text as, if accepted, production will need this reference to link the reader to the figure.

Response: Thank you for pointing out this problem in the manuscript. I made sure that these three images were used in the paper. Figure 1 is on page 7 of the paper. Figure 7 is changed to Figure 5, which is cited on page 24 of the paper. Figure 9 is changed to Figure 7, which is cited on page 27of the paper.

4. We note that Figures 1, 5, 6, 8 and 10 in your submission contain map images which may be copyrighted.

Response: Thank you for pointing out this problem in the manuscript. All my data and pictures are made by myself, and I promise that there is no copyright restriction. At the same time, I also upload all the original data and pictures to the system through " Supporting Information".

5. Please include captions for your Supporting Information files at the end of your manuscript, and update any in-text citations to match accordingly.

Response: Thank you for your rigorous review. We have revised the name of the “supporting information” in accordance with the journal's standards to make it meet the requirements of journal publication. See pages 31-33 of the paper for details.

6. Please upload a copy of Supporting Information Figures and Tables which you refer to in your text on page 33 and 34.

Response: Thank you for your rigorous review. We have sorted out all the data and original pictures in the supporting information, and uploaded them again.

Reviewer #1:

1.The keywords are lack.

Response: We appreciate your valuable comment. We have revised the table of contents of the article based on the PLOS ONE submission template, where the keywords are not directly displayed in the main part of the article but are uploaded separately in the submission system. We present the keywords here and hope to receive your suggestions:

Keywords: land market, residential land price, spatial and temporal evolution, GWR model, driving mechanism

2. Introduction. Too much. Consider transferring part of it to a separate Literature Review. Literature review needs to be integrated, now it is basically simple lists of existing works. “Lack of a comprehensive analysis of the effects of multiple land market factors on land prices,” "seldom consider the spatial relationship between land prices and land market factors" needs to be based on literature evidence. In fact, there must be some literatures discussing multiple factors and considering spatial relationships. Compared with them, what is your innovation or improvement point?

Response: Thank you for your rigorous review. We have integrated the literature in the introduction, and the main contents are structured as follows (See pages 4-6 of the paper for details) : We have reorganized the discussion of the literature in the introduction, especially in paragraphs 3-6, in which paragraphs 3, 4 and 5 introduce the supply and demand mechanism of the land market (land supply), the structure of land supply (primary land market, stock land, guaranteed housing, etc.) and the degree of marketization of land (the proportion of bidding and auction, land offer, etc.) and other hot. Regarding the relationship between area and land price, we have deleted references 16, 17, 26, 32 and integrated other literature to express the information more concisely. Paragraph 6 describes the main models applied to study the influencing factors of land price and the superiority of GWR models in dealing with spatial heterogeneity, and we address the literature on the application of GWR models to study the factors influencing land price. Overall, instead of simply listing the works, the introduction section lays the foundation for the construction of the index system in this paper by summarizing the existing literature on the roles of land market factors in influencing land price, which provided many ideas for the selection of models in this paper.

Differences from other studies: In studies of the spatial relationships between land market factors and land prices, scholars have generally analyzed relationships at the national city level [1-2], with less research conducted at the county level. In terms of the construction of an index system, there are two main approaches: (1) One approach is to construct a comprehensive factor index system to explore the spatial relationship with land price by combining land-use change [9], population structure and distribution [10], regional environment [11-12], urban transportation conditions [13-14], macroeconomic fluctuations [15], and national land policies [16-20]. (2) When exploring the influences of land market factors on land prices, most scholars analyze land supply and demand, land market supply structure, and the degree of land marketization in the direction of modeling with land prices alone and propose specific policy recommendations, while when considering a variety of land market factors in an integrated manner, scholars explore the relationships between the factors and land price without conducting in-depth analyses of spatial heterogeneity [3]; moreover, for the evaluation of maturity within the land market [4-5], there is little connection to land price or house price. In summary, this paper explores the roles of land market factors in driving land prices and spatial variation by constructing a comprehensive land market index system and applying the GWR model.

Yellow-highlighted cited literature is cited in the paper, and the references highlighted in red are as follows:

[1] Feng Yuan, Yehua Dennis Wei, Weiye Xiao. Land marketization, fiscal decentralization, and the dynamics of urban land prices in transitional China[J]. Land Use Policy,2019,89{5}:

[2] Duo Chai, Yunyan Wu, Xiaoping Zhou, Mengrou Lin, Song Zhao. Urban Land Market Differentiation, Land Financial Dependence and Economic Risk Assessment in China: Evidence from 73 Cities′ Land Price Dynamic Monitoring Data[J]. Urban Development Studies. 2018(10) Page:33-40+50

[3] Fan Tu, Jiawei Ge, Daoxue Liu, Qin Zhong. Determinants of Industrial Land Price in the Process of Land Marketization Reform in China[J]. China Land Sciences. 2017(12) Page:33-41+68

[4] Qi Wang, Xiaoyu Ren, Zhongjiang Feng, Jingfeng Ge.Maturity and obstacles of land market in Shijiazhuang city[J]. Journal of Arid Land Resources and Environment. 2019(12) Page:77-82

[5] Zhang Y, Deng C, Xie B, Guoqiang Lei. Development stage evaluation of land markets in Hunan Province based on an entropy weight and matter-element model[J]. Resources Science, 2015, 37(1): 0045-0051.

3.Research area. Per capita GDP and other data should be updated to the latest year.

Response: We thank you for raising this issue. On pages 7 and 8 of the paper, we have updated the socioeconomic data in the study area to the latest year. Since this paper studies the spatial and temporal distribution of residential land prices from 2014-2017, the residential land price data have not been updated.

4. Selection of market indicators for residential land. The selection of indicators needs additional literature support.

Response: Thank you for your rigorous review. In consulting the existing literature, we found that scholars generally focus on the land market price mechanism (as expressed in the relationships between land price and GDP, per capita income, and price level), land supply and demand mechanisms (as expressed in the relationships between land prices and land supply, population and per capita income), land supply structure (as expressed in the relationships between land price and the area of guaranteed housing and the utilization rate of the stock of land), and the degree of land marketization (expressed in the relationships between land price and area offered for sale, land supply rate, and revenue from sale). These four major aspects and land price are modeled and analyzed. The spatial and temporal evolution of land price, as well as the roles and mechanisms of influencing factors, are explored, and policy recommendations are made to maintain stable land prices and the healthy development of the land market.

Adding to the citations of the original literature, we have included citations of the most recent literature on the impacts of various factors in the land market on land prices. See page 12 of the paper for details. The literature is as follows (The literature numbering is consistent with the thesis):

[53] Yuan F, Wei Y D, Xiao W. Land marketization, fiscal decentralization, and the dynamics of urban land prices in transitional China[J]. Land Use Policy, 2019, 89: 104208.

[54] Feng Yuan, Yehua Dennis Wei, Weiye Xiao. Land marketization, fiscal decentralization, and the dynamics of urban land prices in transitional China[J]. Land Use Policy,2019,89(5).

[55] Xiaoyan Shen, Xianjin Huang, Huan Li,Yi Li, Xiaofeng Zhao. Exploring the relationship between urban land supply and housing stock: Evidence from 35 cities in China[J]. Habitat International,2018,77(5).

[56] Hu F Z Y, Qian J. Land-based finance, fiscal autonomy and land supply for affordable housing in urban China: A prefecture-level analysis[J]. Land Use Policy, 2017, 69: 454-460.

[57] Zhang J, Yu T, Li L, Danxia Zhang, Guochao Zhao, Haizhen Wen. Marketization allocation, land price, and local government land speculation, China[J]. International Journal of Strategic Property Management, 2020, 24(5): 335-347.

[58] Odunfa Victoria O, Agboola A. O, Oladokun T. T. Characteristics of Land Market in Nigeria: Case of Ibeju Lekki Local Government, Lagos, Nigeria[J]. Current Urban Studies,2021,09(01).

5. Methods. Explanations and formulas, such as GWR and AIC, can be reduced appropriately, because they are common methods and do not need detailed introduction. This is not the key part.

Response: Thank you for pointing out this problem in the manuscript. To make the article more concise, we have deleted some content regarding the GWR model and have reorganized part of the research methods section. See pages 14 - 16 of the paper for details.

The main changes include the following: 1) We have deleted the content describing the spatial kernel in GWR (including Fig. 2 and Fig. 3; the order of all remaining figures in the paper has been rearranged). 2) In the section describing the optimal bandwidth selection, we have removed the content describing the role of bandwidth variation in the GWR model and have retained the CV method and AIC method formulas. 3) We have cited several articles that provide background information on the GWR model. The cited literature is as follows (The literature numbering is consistent with the thesis):

[59] Xiao L, Lang Y, Christakos G. High-resolution spatiotemporal mapping of PM2. 5 concentrations at Mainland China using a combined BME-GWR technique[J]. Atmospheric Environment, 2018, 173: 295-305.

[60] Gutiérrez-Posada D, Rubiera-Morollon F, Viñuela A. Heterogeneity in the determinants of population growth at the local level: Analysis of the Spanish case with a GWR approach[J]. International Regional Science Review, 2017, 40(3): 211-240.

[61] Xiaoshu C, Jianbin X. Spatial heterogeneity analysis of regional economic development and driving factors in China’s provincial border counties[J]. Acta Geographica Sinica, 2018(06) Page:1065-1075.

6. Results and discussion. Sub-titles. "Analysis of spatial and temporal characteristics of residential land prices" and "Global Spatial Autocorrelation Analysis of Residential Land Prices" are somewhat duplicate, because global spatial autocorrelation belongs to spatial characteristics.

Response: Thank you very much for your careful review. We learned that ESDA and kriging interpolation as well as spatial autocorrelation are essential to explore the spatial and temporal distribution of residential land prices. Therefore, we have restructured the subsections of the results and discussion sections to correspond to the content of the research methods section. We have included "Residential land price ESDA and kriging interpolation analysis" and "Global spatial autocorrelation analysis of residential land price" as subheadings in the section "Spatio-temporal characteristics analysis of residential land price". The results of ESDA and kriging interpolation reveal the characteristics of the spatial and temporal distributions of residential land prices in Beijing, Tianjin, and Hebei; however, we again calculated the global Moran index to further determine the degree of correlation and significance of residential land prices in Beijing, Tianjin, and Hebei. See pages 16 and 19 of the paper for details.

7. The clarity of all images needs to be improved.

Response: We apologize for the inconvenience. We have reprocessed the images in Photoshop for clarity. We apologize again for the unclear images in the paper.

8. The language needs polishing by native speakers to make your work better understood.

Response: We apologize for the language errors in the manuscript and inconvenience they caused. The revised manuscript has been thoroughly edited for English language by a native speaker, so we hope it now meets the journal's standard. Thank you very much for your useful comments. The editing certificate is shown in Figure 1.

Fig 1. Language editing certificate

Reviewer #2:

1. Although the practicality and superiority of the GWR model are described in the introduction section of this paper, the introduction of the GWR model in the research method section is rather messy, and the descriptions of the GWR model in some high-quality papers should be cited to summarize this section to make it concise and clear.

Response: Thank you for pointing out this problem in our manuscript. We noticed that there was too much background information on the GWR model in the research methodology section. Accordingly, we have trimmed down the introduction of the GWR model to make the article more concise. See pages 14-16 of the paper for details

The main changes include the following: 1) We have deleted the content describing the spatial kernel in GWR (including Fig. 2 and Fig. 3; the order of all remaining figures in the paper has been rearranged). 2) In the section describing the optimal bandwidth selection, we have removed the content describing the role of bandwidth variation in the GWR model and have retained the CV method and AIC method formulas. 3) We have cited several articles that provide background information on the GWR model. The cited literature is as follows (The literature numbering is consistent with the thesis):

[59] Xiao L, Lang Y, Christakos G. High-resolution spatiotemporal mapping of PM2. 5 concentrations at Mainland China using a combined BME-GWR technique[J]. Atmospheric Environment, 2018, 173: 295-305.

[60] Xiaoshu C, Jianbin X. Spatial heterogeneity analysis of regional economic development and driving factors in China’s provincial border counties[J]. Acta Geographica Sinica, 2018(06) Page:1065-1075.

[61] Gutiérrez-Posada D, Rubiera-Morollon F, Viñuela A. Heterogeneity in the determinants of population growth at the local level: Analysis of the Spanish case with a GWR approach[J]. International Regional Science Review, 2017, 40(3): 211-240.

2. This paper is relatively clear in explaining the role and mechanisms of positive and negative drivers of land markets. However, the results of the GWR model lack validation, and the results should be further verified by using, for example, Monte Carlo methods.

Response: Thank you for pointing out this problem in the manuscript. We learned that the Monte Carlo method refers to the method of using random numbers to solve many computational problems and can be used to test the significance of the GWR model for the effect of fitting local nonstationarity. We performed Monte Carlo significance tests on the GWR coefficients, and the results are shown in the table below. We can see from the table that the GWR coefficients are all significant, so each influencing factor coefficient has spatial nonstationarity characteristics. We add Monte Carlo tests to the section of the paper on the contribution of land market factors to residential land price changes to make the paper more complete and fluid. Table 1 shows the results of the MC test. See pages 21-22 of the article for details.

Parameter p-value Significance level

Intercept 0.000 ***

X1 0.018 *

X2 0.006 **

X3 0.023 *

X4 0.002 **

X5 0.000 ***

X6 0.001 ***

X7 0.024 *

X8 0.000 ***

X9 0.000 ***

X10 0.000 ***

Note: *** represents the 0.1% significance level; ** represents the 1% wetness level; * represents the 5% significance level.

Table 1.Nonstationarity of the coefficient spatial test based on the Monte Carlo significance test method

3. The conclusion is a good review of the article. However, the policy recommendations need to be specific and should follow the theme of the article, combining the principles of land market indicator system construction, for example, exploring the role of land policy on residential land prices in terms of land market supply and demand, and suggesting corresponding land policy improvements.

Response: Thank you for pointing out this problem in the manuscript. We have reorganized the content of the article and proposed land policy recommendations for the development of the land market, taking into account the roles of land market factors in driving residential land prices and the underlying mechanisms. For details, see pages 31 and 32 of the article; the policy recommendations are reproduced below:

"According to the current market control background of "classification guidance and local policy", different land control policies should be formulated and implemented according to the current situation of supply and demand in the land market in different types of cities. Examples include the following: ① Land supply plans should be formulated and implemented according to the local situation, and the change in land supply should reflect the population size and per capita income, thus guaranteeing a balanced supply and demand in the land market. ② Postsupply supervision of land supply should be strengthened to avoid land idleness, optimize land use structure, excavate the stock of land within the city, and improve land intensification and economization. ③ Optimization of the land market supply method should be continued and the land transfer method should be selected in a scientifically informed manner according to the use purpose of the land transferred to achieve fairness, justice, and openness as much as possible, promote the structural reform of the land supply side, strengthen the proportion of the supply of guaranteed and policy housing, and realize the benign development of the residential market."

4. The introduction section needs to add more up-to-date literature.

Response: Thank you for pointing out this problem in the manuscript. We have cited the latest literature you suggested in the introduction section, see page 3 of the article for details, and the specific information is as follows (The literature numbering is consistent with the thesis):

[2] Zeng P, Sun Z, Chen Y, Qiao S, Cai LW. COVID-19: A Comparative Study of Population Aggregation Patterns in the Central Urban Area of Tianjin, China[J]. International Journal of Environmental Research and Public Health, 2021, 18(4): 2135.

[3] Luo X, Yang J, Sun W, He BJ. Suitability of human settlements in mountainous areas from the perspective of ventilation: A case study of the main urban area of Chongqing[J]. Journal of Cleaner Production, 2021, 310: 127467.

[4] He B J, Ding L, Prasad D. Wind-sensitive urban planning and design: Precinct ventilation performance and its potential for local warming mitigation in an open midrise gridiron precinct[J]. Journal of Building Engineering, 2020, 29: 101145.

[5] Yang J, Wang Y, Xue B, Li YF, Xiao xm, Xia JH. Contribution of urban ventilation to the thermal environment and urban energy demand: Different climate background perspectives[J]. Science of The Total Environment, 2021: 148791.

[6] Yang J, Yang Y, Sun D, Jin C, Xiao xm. Influence of urban morphological characteristics on thermal environment[J]. Sustainable Cities and Society, 2021: 103045.

[7] Yang J, Ren J, Sun D, Xiao xm, Xia JH, Jin C. Understanding land surface temperature impact factors based on local climate zones[J]. Sustainable Cities and Society, 2021, 69: 102818.

5. The language needs to be improved especially the parts of abstract and discussion.

Response: Thank you for your careful review. We apologize for the language errors in the manuscript and inconvenience they caused. The revised manuscript has been thoroughly edited for English language by a native speaker, so we hope it now meets the journal's standard. Figure 2 shows the language editing certificate.

Fig 2. Language editing certificate

---

## [Decision Letter · Decision Letter 1]

13 Aug 2021

Inter-metropolitan land price characteristics and pattern in the Beijing-Tianjin-Hebei urban agglomeration, China

PONE-D-21-20597R1

Dear Dr. an,

We’re pleased to inform you that your manuscript has been judged scientifically suitable for publication and will be formally accepted for publication once it meets all outstanding technical requirements.

An invoice for payment will follow shortly after the formal acceptance. To ensure an efficient process, please log into Editorial Manager at http://www.editorialmanager.com/pone/ , click the 'Update My Information' link at the top of the page, and double check that your user information is up-to-date. If you have any billing related questions, please contact our Author Billing department directly at authorbilling@plos.org.

Kind regards,

Jun Yang

Academic Editor

PLOS ONE

Additional Editor Comments (optional):

Accept

Reviewers' comments:

Reviewer's Responses to Questions

**Comments to the Author**

1. If the authors have adequately addressed your comments raised in a previous round of review and you feel that this manuscript is now acceptable for publication, you may indicate that here to bypass the “Comments to the Author” section, enter your conflict of interest statement in the “Confidential to Editor” section, and submit your "Accept" recommendation.

Reviewer #1: All comments have been addressed

Reviewer #2: All comments have been addressed

2. Is the manuscript technically sound, and do the data support the conclusions?

Reviewer #1: Yes

Reviewer #2: Yes

3. Has the statistical analysis been performed appropriately and rigorously?

Reviewer #1: Yes

Reviewer #2: Yes

4. Have the authors made all data underlying the findings in their manuscript fully available?

Reviewer #1: Yes

Reviewer #2: Yes

5. Is the manuscript presented in an intelligible fashion and written in standard English?

Reviewer #1: Yes

Reviewer #2: Yes

6. Review Comments to the Author

Reviewer #1: The authors have adequately addressed the comments raised in a previous round of review. I consider that this manuscript can now acceptable for publication.

Reviewer #2: The authors have adequately addressed most of the comments raised in a previous round of review and I feel that this manuscript is now acceptable for publication.

7. PLOS authors have the option to publish the peer review history of their article (what does this mean?). If published, this will include your full peer review and any attached files.

Reviewer #1: No

Reviewer #2: No

---

## [Editor Report · Acceptance letter]

18 Aug 2021

PONE-D-21-20597R1

Inter-metropolitan land price characteristics and pattern in the Beijing-Tianjin-Hebei urban agglomeration, China

Dear Dr. Li:

I'm pleased to inform you that your manuscript has been deemed suitable for publication in PLOS ONE. Congratulations! Your manuscript is now with our production department.

Kind regards,

on behalf of

Dr. Jun Yang

Academic Editor

PLOS ONE